# Neutralizing IFNγ improves safety without compromising efficacy of CAR-T cell therapy in B-cell malignancies

Simona Manni [1,5], Francesca Del Bufalo [1,5], Pietro Merli[1], Domenico Alessandro Silvestris[1], Marika Guercio[1], Simona Caruso[1], Sofia Reddel[1], Laura Iaffaldano[1], Michele Pezzella [1], Stefano Di Cecca[1], Matilde Sinibaldi[1], Alessio Ottaviani [1], Maria Cecilia Quadraccia [1], Mariasole Aurigemma [1], Andrea Sarcinelli [1], Roselia Ciccone[1], Zeinab Abbaszadeh[1], Manuela Ceccarelli [1], Rita De Vito[2], Maria Chiara Lodi[1], Maria Giuseppina Cefalo[1], Angela Mastronuzzi [1], Biagio De Angelis [1] ✉, Franco Locatelli [1,3,5] ✉ & Concetta Quintarelli[1,4,5]

Chimeric antigen receptor T (CAR-T) cell therapy may achieve long-lasting remission in patients with B-cell malignancies not responding to conventional therapies. However, potentially severe and hard-to-manage side effects, including cytokine release syndrome (CRS), neurotoxicity and macrophage activation syndrome, and the lack of pathophysiological experimental models limit the applicability and development of this form of therapy. Here we present a comprehensive humanized mouse model, by which we show that IFNγ neutralization by the clinically approved monoclonal antibody, emapalumab, mitigates severe toxicity related to CAR-T cell therapy. We demonstrate that emapalumab reduces the pro-inflammatory environment in the model, thus allowing control of severe CRS and preventing brain damage, characterized by multifocal hemorrhages. Importantly, our in vitro and in vivo experiments show that IFNγ inhibition does not affect the ability of CD19-targeting CAR-T (CAR.CD19-T) cells to eradicate CD19+ lymphoma cells. Thus, our study provides evidence that anti-IFNγ treatment might reduce immune related adverse effect without compromising therapeutic success and provides rationale for an emapalumab-CAR.CD19-T cell combination therapy in humans.

Adoptive cellular immunotherapy with chimeric antigen receptor T (CAR-T) cells represents a revolutionary approach for the treatment of B-cell lymphoproliferative disorders either relapsed or refractory to conventional therapies, including hematopoietic stem cell (HSC) transplantation. This innovative treatment, however, is characterized by peculiar toxicities. In particular, cytokine release syndrome (CRS)

and immune effector cell-associated neurotoxicity syndrome (ICANS) have been largely reported in treated patients, with severity ranging from mild symptoms to life-threatening events or even death[1]. In addition, as a manifestation of severe CRS or as its pathophysiological consequence, the occurrence of hemophagocytic lymphohistiocytosis (HLH)/macrophage activation syndrome (MAS) has been reported in

[1]Department of Haematology-Oncology and Cell and Gene Therapy, Bambino Gesù Children Hospital, IRCCS, Rome, Italy. [2]Department of Pathological Anatomy, Bambino Gesù Children Hospital, IRCCS, Rome, Italy. [3]Department of Pediatrics, Catholic University of the Sacred Heart, Rome, Italy. [4]Department of Clinical Medicine and Surgery, University of Naples Federico II, Naples, Italy. [5]These authors contributed equally: Simona Manni, Francesca Del Bufalo, Franco Locatelli, Concetta Quintarelli. ✉e-mail: biagio.deangelis@opbg.net; franco.locatelli@opbg.net

patients treated with CAR.CD19[2,3] or CAR.CD22-T cells[4]. HLH/MAS are clinical syndromes characterized by pathological hyperinflammation and uncontrolled macrophage activation, with symptoms largely overlapping those of CRS. Thus, CRS and HLH/MAS belong to a spectrum of systemic hyperinflammatory disorders. Notably, the recently published guidelines define this complication as a CRS/MAS overlap syndrome, typically characterized by persistent fever despite administration of the anti-IL-6 receptor antibody, tocilizumab, organomegaly, cytopenias (±hemophagocytosis in the bone marrow), hyperferritinemia (>10,000 ng/ml), liver dysfunction, coagulopathy (characterized, in particular, by hypofibrinogenemia) and hypertrigliceridemia[5].

CRS typically occurs within a few days after CAR-T cell infusion, often concomitantly with the maximal CAR-T cell expansion in vivo[6]. Notably, CRS incidence has been reported to range between 30 and 100%, with 10–30% of cases being either severe or life-threatening[7,8]. ICANS, a neurological complication characterized by irritability, dizziness, and disorientation in mild forms, until global severe encephalopathy in the most severe cases, is reported to occur in 20–67% of lymphoma patients[9], and in 29–72% of patients with B-cell precursor acute lymphoblastic leukemia (Bcp-ALL)[10]. It can occur concurrently with CRS, slightly after CRS or following its resolution; approximately 10% of patients develop a form of "delayed ICAN" occurring >3 weeks after infusion[11,12].

Although the pathophysiology of ICANS remains less understood than that of CRS, both toxicities are characterized by a systemic inflammatory cascade with massive production of cytokines following the binding of CAR-T cell receptor to its target antigen. Upon activation of CAR-T cells, there is a release of high levels of IFNγ, whose ability to induce the activation of other immune cell subsets, mostly macrophages, is well known[13]. This inflammatory loop, involving activated CAR-T cells, lysed cancer cells and macrophages, results into the massive production of additional cytokines, including IL-6, TNF-α, and IL-10, together with several catecholamines[14], exponentially amplifying the inflammatory response. This process, on one hand, increases the anti-leukemia activity of CAR-T cells[14], but on the other, it can evolve into an uncontrolled, harmful condition.

While low-grade CRS is usually easily managed using symptomatic measures and supportive care and, if needed, tocilizumab, high-grade CRS requires the administration of steroids after tocilizumab to control the inflammatory cascade before the occurrence of organ failures[6]. Unfortunately, it has been reported that the prolonged use of high-dose steroids affects the antitumor activity of CAR-T cells, impairing the long-term outcome of patients[15,16]. Moreover, the use of tocilizumab has some limitations. In particular, several groups found that early intervention with tocilizumab largely mitigates the risk of sCRS, but does not affect either most cases of fulminant refractory HLH/MAS[17], or treatment of ICANS[18]. This inefficacy of ICANS may be related to the different pathophysiology between CRS and ICANS, as well as to the poor penetration of tocilizumab across the blood–brain barrier (BBB). Indeed, it has been demonstrated that prophylactic use of tocilizumab decreases the incidence of sCRS, but increases the occurrence of severe ICANS[6,11], likely owing to the higher circulating levels of IL-6 induced by the receptor blockade that, in turn, results in higher IL-6 levels in the central nervous system (CNS)[19–21]. In view of these findings, the most recent guidelines recommend the use of corticosteroid therapy for grade ≥2 ICANS[5].

As already mentioned, high levels of IFNγ have been found in patients with CRS, supporting the hypothesis that this cytokine plays a pivotal role in driving the inflammatory process[15]. Notably, pericytes exposed to IFNγ contribute to the activation of endothelial cells and to the increase of BBB permeability, a dysfunction that appears to be a relevant finding in neurotoxicity: after the occurrence of BBB disruption, IFNγ and TNF-α can activate the immune effector microglia cells,

whose cytokines, released upon activation, induce neuroinflammation and brain damage[22].

In view of all these considerations, we evaluate whether inhibition of the IFNγ axis through emapalumab, a human monoclonal antibody (mAb) directed against human IFNγ already approved by the FDA since November 2018 for treatment of refractory, relapsed primary HLH (pHLH)[23], can offer a therapeutic opportunity to manage these complications. Our results indicate that emapalumab is able to induce a significant reduction of CRS, HLH/MAS and brain damage, without impairing the anti-leukemia activity of CAR-T cells against CD19-positive B-cell malignancies. Preclinical data from this paper offer the rational for the design of clinical trials based on the use of emapalumab with the aim to control CAR-T cell-associated toxicity.

## Results

### Neutralization of IFNγ does not impair CAR.CD19-T cell anti-leukemic capacity in vitro

In order to obtain preliminary evidence of the clinical suitability of emapalumab to manage CAR-T cell toxicities, we first evaluated whether neutralization of the IFNγ axis affects the cytotoxic activity CAR-T cells against CD19+ tumor cells. For this purpose, peripheral blood mononuclear cells (PBMCs) derived from healthy donor (HD) were genetically modified with the 4-1BB-based CAR.CD19 construct, which was used clinically at the Bambino Gesù Children Hospital of Rome (NCT03373071). In detail, the anti-lymphoma activity of CAR.CD19-T cells was tested in long-term co-culture assays with CD19+ lymphoma Daudi (Fig. 1a) and Raji (Supplementary Fig. 1a) cell lines, either in the absence or in the presence of the anti-IFNγ mAb emapalumab. The concentration of emapalumab was in the range of 0.2–100 µg/ml. This dose range corresponds to the concentration of the drug observed in the plasma of individual healthy subjects receiving emapalumab (study NI-0501-03)[24]. The presence of emapalumab in the culture did not cause any significant modification of the cytotoxicity exerted by CAR.CD19-T cells toward CD19+ lymphoma cells (Effector:Target, E:T ratio of 1:1) in vitro, as compared to untreated conditions, at any of the tested concentrations (Fig. 1a for Daudi model and Supplementary Fig. 1a for Raji model). The evaluation of cytokines in the supernatant of the co-culture confirmed that IFNγ was completely neutralized at the doses used (Fig. 1b and Supplementary Fig. 1b; $p < 0.05$ for the tested conditions), whereas no significant differences were observed for Granzyme B, IL-2 and TNFα. In order to confirm these results in a more stressed condition, which could reproduce the clinical situation more reliably, the same experiment of co-culture with lymphoma cells was also performed at suboptimal conditions of decreased E:T ratio (from 1:1 to 1:80), to further evaluate whether the IFNγ neutralization could impact the cytotoxic activity of CAR.CD19-T cells. In these experiments, emapalumab was added to the co-cultures at the highest dose of 100 µg/ml. As shown in Fig. 1c and Supplementary Fig. 1c, the anti-lymphoma activity of CAR.CD19-T cells was superimposable to the control condition, in which emapalumab was not added to the co-culture, at all the ratios tested.

Furthermore, to test if the IFNγ neutralization interferes with the kinetics of target killing in short (few hours) and long timeframes (up to 64 h), the anti-lymphoma activity of CAR.CD19-T cells against Daudi tumor cells was analyzed through a real-time, quantitative, live-cell imaging system that allows the visualization and quantification of live GFP+ cells over time. As shown in Fig. 1d, CAR.CD19-T cells exposed to emapalumab (dotted black line) exhibit a cytotoxicity profile superimposable to that of the untreated CAR.CD19-T cells (black line). Moreover, we did not find differences in the proliferation index of CAR.CD19-T cells induced by the 72 h exposure to Daudi cells either in the absence (Fig. 1e, blue area) or in the presence of 100 µg/ml emapalumab (Fig. 1e, purple area). Lastly, we analyzed CAR-T cell cytotoxicity at a very early time-point by measuring the bioluminescence of Raji cells, genetically modified to express firefly luciferase (FF-Luc),

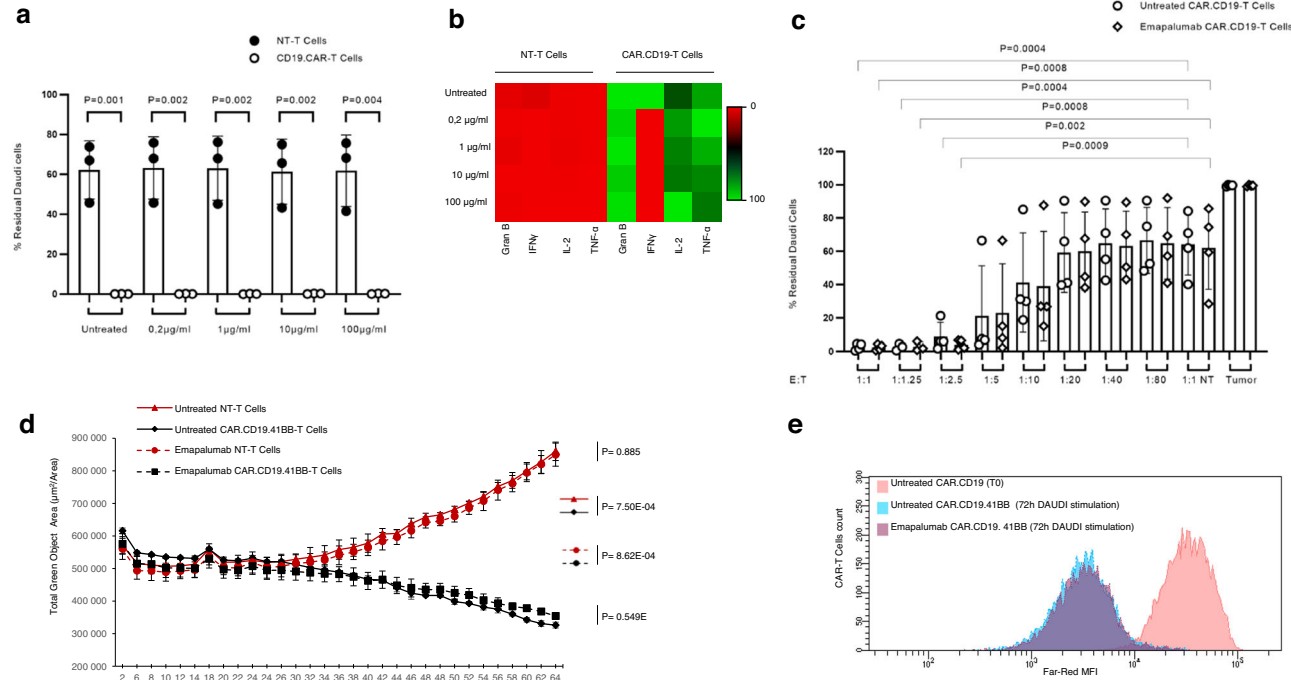

**Fig. 1 | Emapalumab does not affect CAR.CD19-T cell in vitro cytotoxicity against Daudi tumor cells. a** NT or CAR.CD19.4-1BB-T cells and GFP+ Daudi cells were plated at the 1:1 E:T ratio in the presence or absence of emapalumab within the concentration range of 0.2–100 µg/ml. After 7 days, FACS analysis was performed to detect GFP+ residual tumor. Data are expressed in percentage, as mean ± SD of 3 independent experiments, with effector cells generated from 3 HDs. Data were compared by a two-tailed Student *t*-test and *p* value <0.05 was considered statistically significant. **b** Heatmaps displaying the IFNγ, Granzyme B, IL-2 and TNF-α levels that are secreted by NT or CAR.CD19.4-1BB-T cells after 24 h of co-culture with Daudi cells in media containing emapalumab within the concentration range of 0–100 µg/ml. Color scale represents cytokine levels as mean of 3 independent experiments with effector cells generated from 3 HDs expressed in percentage. **c** NT or CAR.CD19.4-1BB-T cells and GFP+ Daudi cells were plated at decreasing E:T ratios, from 1:1 to 1:80 either in the presence or in the absence of 100 µg/ml emapalumab. After 7 days, FACS analysis was performed to detect GFP+ residual tumor cells. Data are expressed in percentage, as mean ± SD of 3 independent

experiments with effector cells generated from 3 HDs. Data were compared by a two-tailed Student *t*-test and *p* value <0.05 was considered statistically significant. **d** Real-time kinetics of Daudi elimination exerted by T cells (NT or CAR.CD19.4-1BB) (1:1 E:T ratio) either in the presence or absence of 100 µg/ml emapalumab. The indicated statistical significance refers to the end point of the assay. Data are expressed as mean ± SD of 3 replicates. Data were compared by a two-tailed Student *t*-test and *p* value <0.05 was considered statistically significant. **e** Proliferation of CAR.CD19.4-1BB-T cells activated by Daudi cells and monitored for 3 days in the absence (blue area) or in the presence of 100 µg/ml emapalumab (purple area) compared to CAR-T cells at time of cell plating (day 0, pink area). Exemplificative histograms, relative to one representative experiment, out of the three performed with three different CAR.CD19-T cell donors (*p* value of the proportion of proliferating CAR.CD19-T cells with vs. without Emapalumab: 0.983). Data are expressed as mean ± SD. Significant *p* value <0.05. Source data are provided as a Source Data file.

after 6 h of co-culture with CAR.CD19-T cells, at the E:T ratio 1:1. As shown in Supplementary Fig. 1d, we did not observe any significant decrease in the elimination index in the presence of emapalumab. These data were further corroborated by the evaluation, by flow-cytometry, of the markers of apoptosis (Annexin V and 7-AAD) on Raji cells, after 6 h of co-culture with CAR.CD19-T cells at a 1:1 E:T ratio. As shown in Supplementary Fig. 1e, the IFNγ neutralization did not impact significantly the percentage of early apoptotic Raji cells.

Several studies have reported that CAR-T cells incorporating 4-1BB costimulatory domains have a significant differential signature and kinetics of lymphoma elimination as compared to CAR-T cells characterized by the presence of the CD28 costimulatory domain[25]. We thus investigated whether the antitumor activity of CAR.CD19.CD28-T cells could be affected by the presence of emapalumab. As shown in Fig. 2a, CAR.CD19.CD28-T cells were co-cultured with Daudi tumor cells at suboptimal conditions of decreased E:T ratio (from 1:1 to 1:2.5), showing that, also in this case, the antitumor activity of CAR.CD19-T cells exposed to emapalumab at the highest dose of 100 µg/ml was superimposable to that of CAR-T cells in the absence of the drug.

Moreover, the evaluation of cytokines in the supernatant collected after 24 h of co-culture confirmed that IFNγ was completely neutralized (Fig. 2b), whereas no significant differences were observed for Granzyme B, IL-2 and TNFα (Fig. 2b). Furthermore, the killing

activity of CAR.CD19-T cells against Daudi cells analyzed through the real-time imaging system (Fig. 2c), as well as the proliferation assay (Fig. 2d), reveal that IFNγ neutralization does not affect the functionality of CD28-based CAR.CD19-T cells in either a short or long-time frame experimental setting.

## Emapalumab does not impair CAR.CD19-T cell activation signaling

In order to investigate if IFNγ neutralization could have an impact on the CAR-T cell gene signature, we analyzed the expression of 780 genes associated with some essential pathways of CAR-T biology, including phenotype, cell types, exhaustion, metabolic fitness, TCR diversity, toxicity, activation and persistence, in activated CAR.CD19-T cells, with or without exposure to emapalumab. In detail, non-transduced T (NT-T) cells and CAR.CD19-T cells were activated with 0.5 µg/ml recombinant human CD19 Fc Chimera Protein for 16 h before the analysis of the gene signature. Overall, 90 genes resulted to be differentially regulated in the CAR.CD19-T cell samples as compared to the control NT cells (46 upregulated and 44 downregulated genes), all involved in activation pathways of T cells (Supplementary Fig. 2a, b and Supplementary Data 1). Upon addition of emapalumab (100 µg/ml) to activated CAR.CD19-T cells, we observed a differential expression of 14 genes (Fig. 3a, b). Among the 14 deregulated genes,

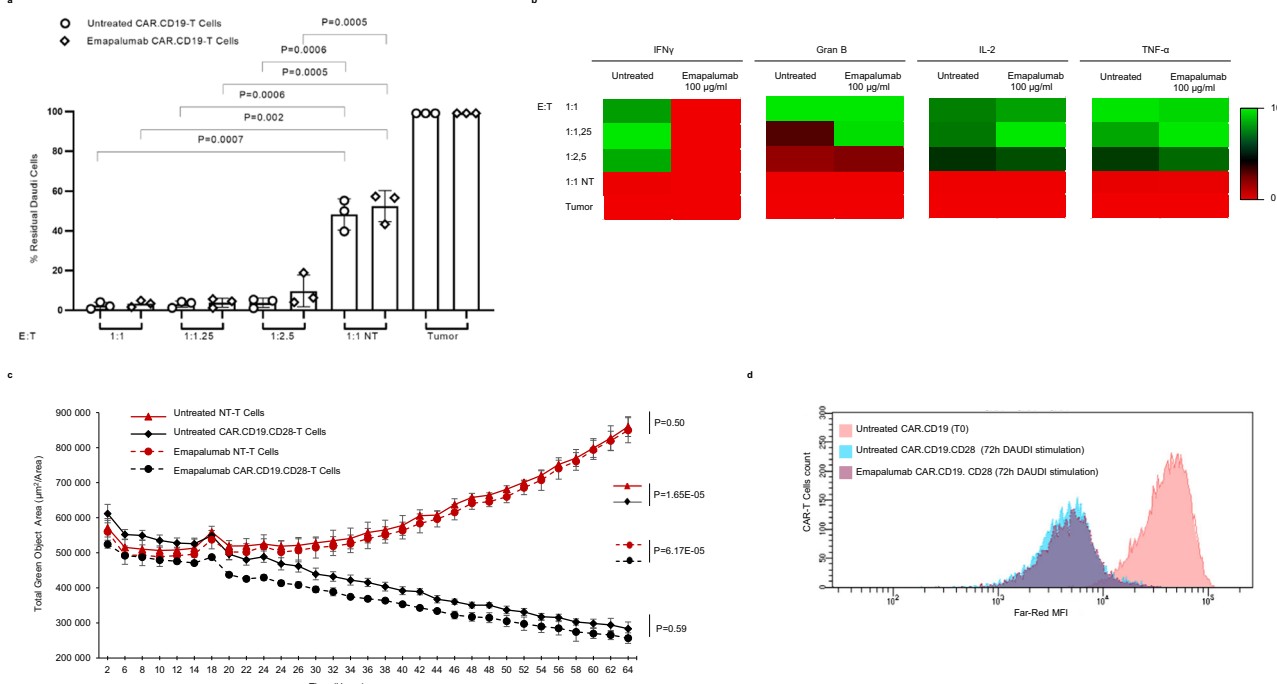

**Fig. 2 | Emapalumab does not affect CD28-based CAR.CD19-T cell in vitro cytotoxicity. a** Effector T cells (NT or CAR.CD19.CD28) and GFP+ Daudi cells were plated at decreasing E:T ratios, from 1:1 to 1:2.5 either in the presence or absence of 100 µg/ml emapalumab. At the end of the experiments, tumor cells and T cells were collected and assessed by FACS analysis evaluating the residual GFP+ tumor cells. Data are expressed in percentage, as mean ± SD of 3 independent experiments, with effector cells generated from 3 HDs. Data were compared by a two-tailed Student *t*-test and *p* value <0.05 was considered statistically significant. **b** Heatmaps displaying the IFNγ, Granzyme B, IL-2 and TNF-α levels that are secreted by effector T cells (NT or CD28-based CAR.CD19) after 24 h of co-culture with Daudi cells in media containing emapalumab within the concentration range of 0.2–100 µg/ml. Color scale represents cytokine levels as mean of 3 independent experiments with effector cells generated from 3 HDs expressed in percentage. **c** Real-time kinetics of Daudi elimination exerted by T cells (NT or

CAR.CD19.CD28) (1:1 E:T ratio) either in the presence or absence of 100 µg/ml emapalumab. The indicated statistical significance refers to the end point of the experimental assay. Data are expressed as mean ± SD of 3 replicates. Data were compared by a two-tailed Student *t*-test and *p* value <0.05 was considered statistically significant. **d** Proliferation of CAR.CD19.CD28-T cells activated by Daudi cells and monitored for 3 days in the absence (blue area) or in the presence of 100 µg/ml emapalumab (purple area) compared to CAR-T cells at the time of cell plating (day 0, pink area). Exemplificative histograms, relative to one representative experiment, out of the three performed with three different CAR.CD19-T cell donors (*p* value of the proportion of proliferating CAR.CD19-T cells with vs. without Emapalumab: 0.957). Student's *t*-test was employed to calculate statistically significant differences between groups. Source data are provided as a Source Data file.

the only upregulated is ITGAM, a gene associated with persistence and T-cell migration signaling (Supplementary Table 1). All the other 13 downregulated genes, as expected, are involved in Interferon signaling pathways, or have already been reported as genes associated with toxicity (gene signature has been evaluated by bioinformatics analysis through PubMed database, Supplementary Table 1), and the majority of them characterized by a linked signaling (Fig. 3c and Supplementary Data 2). Moreover, seven out of the 14 genes deregulated by IFNγ neutralization, were specifically upregulated by the CAR.CD19 stimulation (Fig. 3d, e).

Lastly, to confirm the data on gene expression with the phenotypic characterization of CAR.CD19-T cells, the CAR.CD19-T cell activation phenotype was tested upon challenge with lymphoma cells (E:T ratio of 1:1), in the presence or absence of emapalumab. As shown in Fig. 4, we corroborated the gene expression data by monitoring the expression of several activation markers, including CD25, CD40L, HLA-DR, CD28, CD38, CD44 and CD69, showing no significant modulation when CAR.CD19-T cells are stimulated by the lymphoma DAUDI cells in the presence of INFγ neutralization.

## IFNγ neutralization preserves CAR.CD19-T cell anti-lymphoma activity in a murine in vivo model

We then tested the impact of emapalumab on the anti-lymphoma activity of CAR.CD19-T cells in the in vivo setting, in two different treatment settings.

First, we considered an animal model in which CAR.CD19-T cells were infused when the tumor burden was low, mimicking most of the clinical applications of CAR-T cells in Bcp-ALL patients. In this setting, clinical practice has shown that the onset of toxicity is delayed and both the incidence and severity of toxicity are limited[26]. In light of these considerations, we administered emapalumab 7 days after CAR-T cell infusion (Fig. 5a shows the experimental design of the in vivo model). Mice were intravenously (i.v.) engrafted with Daudi FF-LUC cells on day −2 and tumor burden was then monitored by in vivo imaging system (IVIS). After tumor engraftment, mice received i.v. control NT or 10 × 10⁶ CAR.CD19-T cells, followed by the intraperitoneal (i.p.) administration of either control vehicle or 100 mg/kg emapalumab on Day 7, 11, 15, corresponding to the doses used for the non-clinical in vivo pharmacology/toxicology studies of emapalumab. [https://www.fda.gov/drugs/fda-approves-emapalumab-hemophagocytic-lymphohistiocytosis]. Notably, tumor was still largely detectable on the first day of emapalumab administration (Day 7).

As shown in Fig. 5b, c, CAR.CD19-T cells in the presence of emapalumab exerted a significant lymphoma control (Fig. 5b, mice from #13 to #16; Fig. 5c, d, bioluminescence average value at day 28 = 5.87E + 05 ± 8.17E + 04), which did not differ from that of mice not exposed to emapalumab (Fig. 5b, mice from #9 to #12; Fig. 5c, bioluminescence average value at day 28 = 6.18E + 05 ± 9.59E + 04; *p* = 0.58).

In the second animal model, we wanted to mimic the clinical situation of a high tumor burden, often associated with acute toxicity

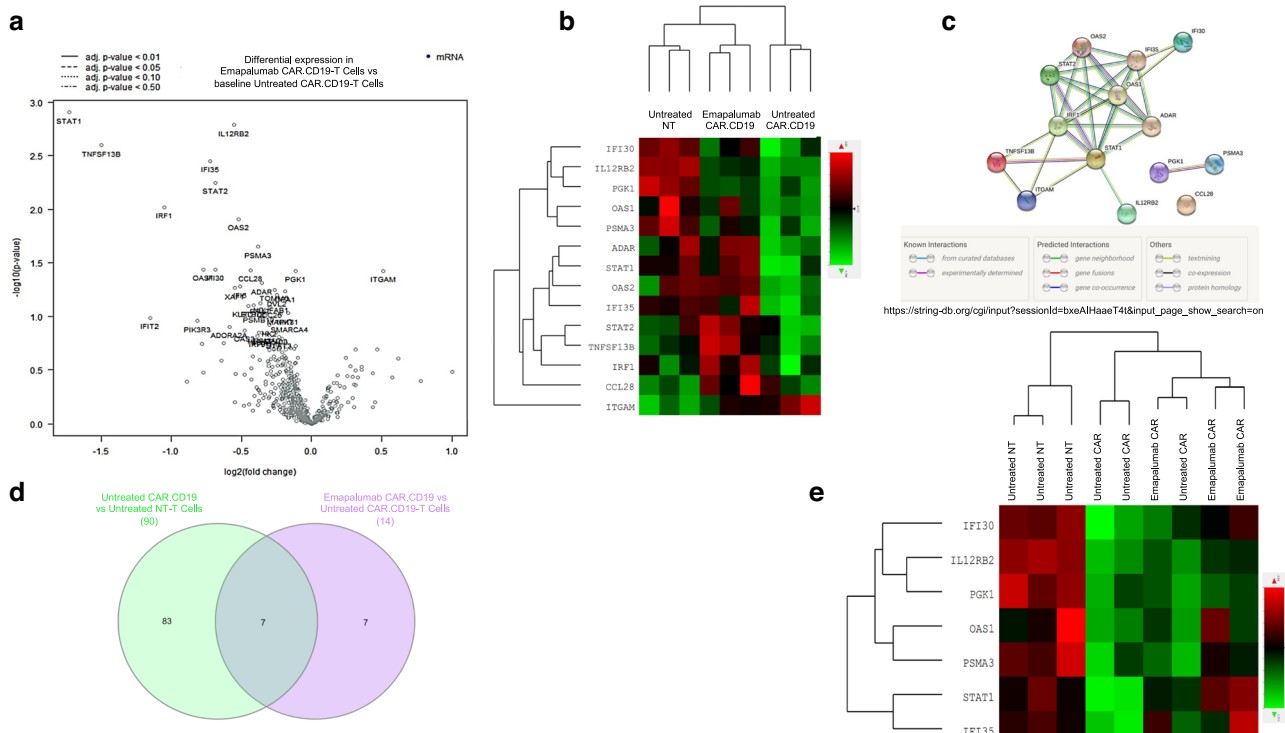

**Fig. 3 | IFNγ neutralization does not impair CAR.CD19-T cell activation signaling. a** Volcano plot of the 14 deregulated genes in CAR.CD19-T cells activated with 0.5 µg/ml Recombinant Human CD19 Fc Chimera Protein for 16 h in the absence or presence of 100 µg/ml emapalumab. The log2FC indicates the mean expression level for each gene. Genes with negative log2FC are downregulated in CAR.CD19-T cells in the presence of emapalumab. **b** Heatmap displaying the expression of the same genes in untreated NT-T cells, untreated CAR.CD19-T cells and CAR.CD19-T cells treated with 100 µg/ml emapalumab. **c** STRING tool has been applied to deconvolute the potential interaction network among the 14 genes deregulated in CAR.CD19-T cells in the presence of emapalumab. **d** Venn diagram that summarizes common genes deregulated both in untreated CAR.CD19 compared to untreated NT-T cells and in CAR.CD19 cultured either in the presence or absence of emapalumab (untreated CAR.CD19-T cells). **e** Heatmaps displaying the 7 genes specifically upregulated in activated CAR.CD19-T cells, but deregulated in the presence of emapalumab. Data are obtained from three independent experiments, with effector cells generated from 3 HDs. Source data are provided as a Source Data file.

in patients treated with CAR.CD19-T cells[26]. In this case, the administration of emapalumab was considered in a schedule of preventive treatment. In particular, mice were i.v. engrafted with Daudi FF-LUC cells on day −8 to allow a tumor burden to grow up to $10^7$ level of bioluminescence (s/cm$^2$/sr) before the i.v. infusion of either NT or CAR.CD19-T cells ($10 \times 10^6$ cells per mouse) at day 0 (Fig. 5e shows the experimental design of the second in vivo model). Emapalumab (100 mg/kg) was administered i.p. on Day 0, 3, 6. Also in this second animal model, we were able to provide evidence that CAR.CD19-T cells exerted a significant lymphoma control in the presence of IFNγ neutralization (Fig. 5f, mice #13 to #16; Fig. 5g, bioluminescence values—dotted black lines—for each animal over time; Fig. 5h, bioluminescence average value at day 15 = 5.67E + 07; $p$ = 0.0002 vs. the cohort of mice receiving NT-T cells), which did not differ from that of the untreated counterpart (Fig. 5f, mice #9 to #12; Fig. 5g, bioluminescence values—black lines—for each animal over time; Fig. 5h, bioluminescence average value at day 15 = 2.18E + 07; $p$ = 0.47).

As shown in Supplementary Fig. 3a, in this latter setting, we have also monitored the IFNγ neutralization that persists until the end of the in vivo experiment, without affecting the in vivo CAR.CD19-T cell expansion (Supplementary Fig. 3b–e).

This in vivo study was also performed using lower doses of CAR.CD19-T cells to investigate whether emapalumab administration affects CAR-T cell activity and proliferation in a more stringent and challenging setting characterized by a low CAR-T cell dose. For this reason, after tumor engraftment, mice received i.v. infusion of $1 \times 10^6$ or $0.1 \times 10^6$ CAR.CD19-T cells, followed by the i.p. administration of 100 mg/ kg of emapalumab or control vehicle on days 0, 3 and 6 (experimental setting in Fig. 5e, control NT-T cell conditions showed in Fig. 5f).

We did not observe any difference in the bioluminescence between emapalumab-treated or untreated mice for both CAR-T cell doses we used (Supplementary Fig. 4a, b), with, in particular, significant control of the lymphoma tumor being only observed in the cohort of mice receiving $1 \times 10^6$ CAR-T cells and emapalumab (Supplementary Fig. 4a, b).

## Development of a simplified humanized murine model effectively reproducing CRS and brain damage

hGM-CSF/hIL3 NOG mice engrafted with human CD34+ HSCs were used in order to establish an animal model recapitulating the human CRS and brain damage. hGM-CSF/hIL3 NOG mice, indeed, stably develop extensive human myeloid and lymphoid cell lineages which are present in peripheral blood, bone marrow, thymus, spleen and non-lymphoid tissue including lung and liver. Engraftment of human hematopoietic CD45$^+$ cells in the peripheral blood (Supplementary Fig. 5a), whose distribution was characterized by the presence of B cells, T cells and CD14+ cells (Supplementary Fig. 5b), was confirmed before proceeding with the experimental plan. Notably, in order to provide a non-proliferating tumor substrate, aimed exclusively to activate CAR-T cells and to avoid developing uncontrolled tumor burden, thus confounding the CRS symptoms and impacting mouse overall survival, humanized hGM-CSF/hIL3 NOG mice were engrafted with irradiated CD19+ Daudi (irrDaudi) FF-LUC cell lines by i.v. injection (Fig. 6a). After the infusion of Daudi cells, mice received i.v. infusion of CAR.CD19-T cells ($10 \times 10^6$/mouse), generated from HDs. CRS signs, including general suffering, as well as circulating proinflammatory cytokines, were monitored over the course of 15 days. We observed that humanized mice infused with irrDaudi and CAR-T

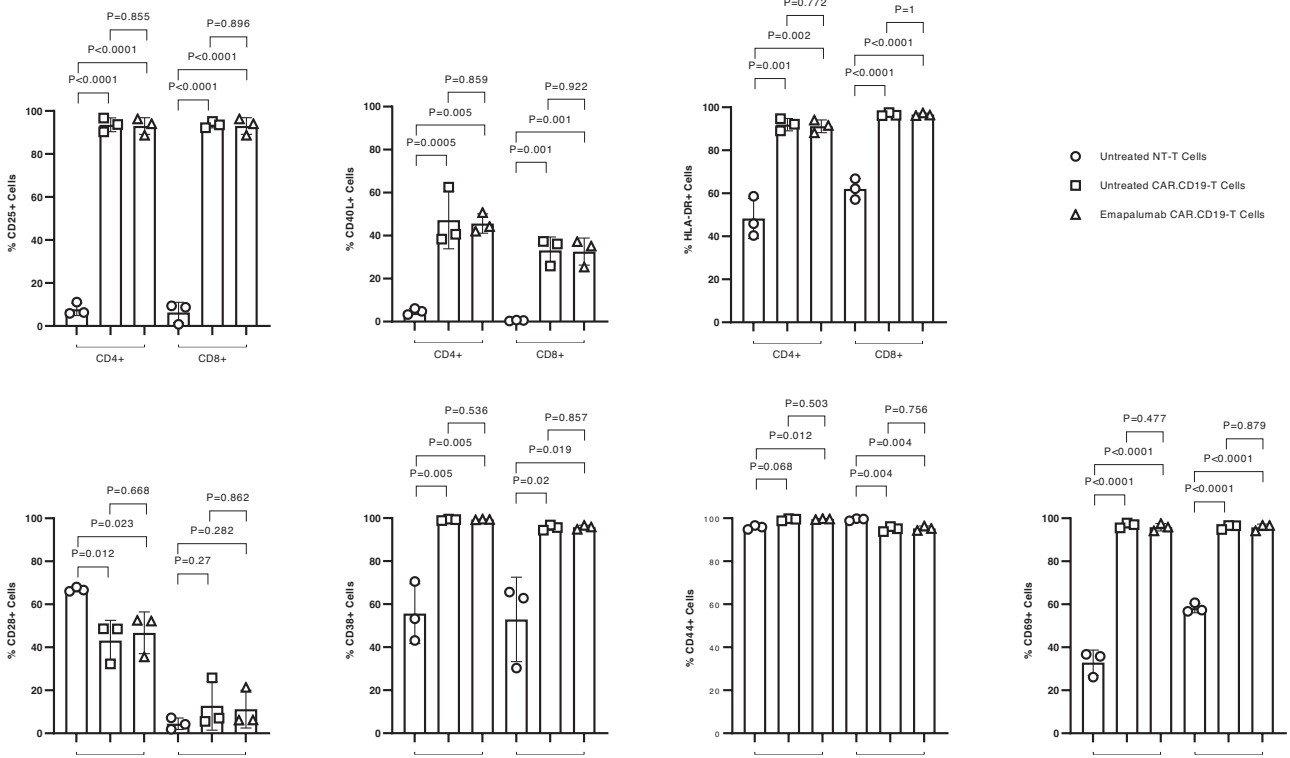

**Fig. 4 | IFNγ neutralization does not affect the activation phenotypic profile of CAR.CD19-T cells.** Phenotypic analysis on different activation markers expressed by T cells (NT or CAR.CD19) after 16 h of co-cultures with Daudi cells either in the presence or absence of 100 µg/ml emapalumab. Data are expressed as mean ± SD and obtained by three independent experiments, with CAR.CD19-T cells generated by three different donors. Student's *t*-test was employed to evaluate the statistical relevance of the differences between groups. Data were compared by a two-tailed Student *t*-test and *p* value <0.05 was considered statistically significant. Source data are provided as a Source Data file.

cells showed a significant increase of pro-inflammatory cytokines, including IFNγ (Fig. 6b), IL-6 (Fig. 6c), TNF-α (Supplementary Fig. 6b), the INFγ-driven chemokine CXCL9 (Fig. 6d), CXCL10 (Fig. 6e) as well as IL-1β (Supplementary Fig. 6a), although this last detected to a low level of 3.96 ± 2.8 pg/ml.

Notably, this CRS model was characterized by severe toxicity, with all mice dying in the first 7 days after CAR.CD19-T cell infusion, without the potentially confounding signs of sufferance due to lymphoma cell expansion, since the mice were engrafted with irradiated not-proliferating lymphoma cells (Fig. 6f). In addition, considering the severity of the generated model, an ad-hoc experiment was designed to evaluate the brain damage occurrence in this murine model of acute CAR-T cell toxicity (Fig. 7a), by sacrificing mice 3 days after CAR-T cell infusion. In this setting, the three doses of emapalumab were administered daily (Fig. 7a). In mice given both irrDaudi and CAR.CD19-T cells, we observed a significant increase in the occurrence of hemorrhagic areas in the CNS of mice developing CRS, as compared to control mice infused only with irrDaudi (Fig. 7b, c; *p* = 0.035).

To study the relevance of our experimental model for the occurrence of HLH/MAS, we analyzed bone marrow samples obtained from mouse tibiae. We observed a marked "starry sky" appearance, representing phagocytes that have engulfed apoptotic cells (Supplementary Fig. 7, middle panels). HLH/MAS histological findings were never observed in control mice infused only with irrDaudi (Supplementary Fig. 7, top panels).

### Emapalumab is highly effective in controlling acute and fatal toxicity related to CAR.CD19-T cells
The humanized model was instrumental to study the ability of emapalumab to control CAR.CD19-T cell toxicity. As shown in Fig. 6b,

emapalumab infusion was associated with a significant reduction in the serum levels of the INFγ-driven chemokine CXCL9 (Fig. 6d; *p* = 0.007 at Day+3 and *p* = 0.01 at Day+4), CXCL10 (Fig. 6e; *p* = 0.002 at Day+3 and *p* = 0.005 at Day+4) and IL-6 (Fig. 6c; *p* = 0.009 at Day+3 and *p* = 0.01 at Day+4), whereas had no impact on the percentage of detected CAR-T cells in treated mice compared to mice that did not receive the drug (Supplementary Fig. 8). Most importantly, mice engrafted with irrDaudi and treated with the concomitant infusion of CAR.CD19-T cells and emapalumab showed an overall survival of 100%, even extending the follow-up to 15 days (Fig. 6f, *p* = 0.02 vs. control untreated mice). Moreover, in the animals receiving CAR-T cells and emapalumab, we observed a significant reduction of the hemorrhagic areas in brains compared to mice not given emapalumab, their histology being similar to that observed in control animals that did not receive CAR-T cells (Fig. 7b, c; *p* = 0.045). To deeply evaluate the impact of IFNγ neutralization in reducing brain damage in mice developing CRS, brain tissue sections were analyzed for the expression of 770 human genes involved in the cellular stress and injury response, glial regulatory pathways, inflammation and peripheral immune invasion, glial cell homeostasis and activation, and neurotransmission. As shown in Fig. 8a and Supplementary Table 2, 32 genes were significantly downregulated in emapalumab-treated mice compared to their untreated counterparts. Notably, these genes are characterized by linked signaling (Fig. 8), significantly associated with biological pathways relevant to our setting, namely oxidative stress, IFNγ signaling, chemokine- and cytokine-mediated inflammation (Fig. 8c).

Lastly, histopathologic evaluation of tibia bone marrow from mice infused with CAR.CD19-T cells and treated with emapalumab showed a significant reduction in phagocytic cells or mitoses (Supplementary

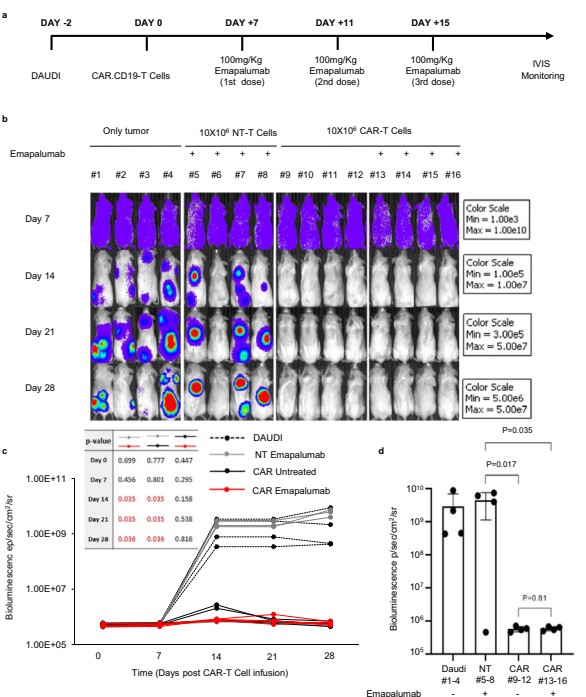

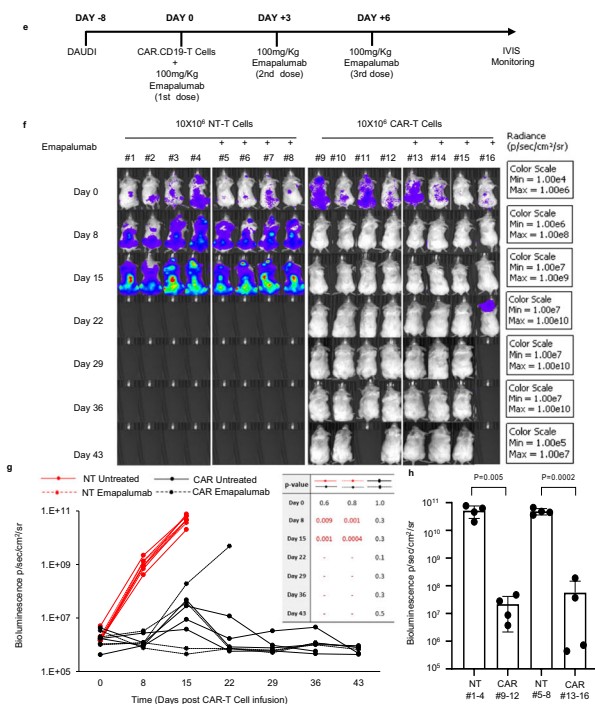

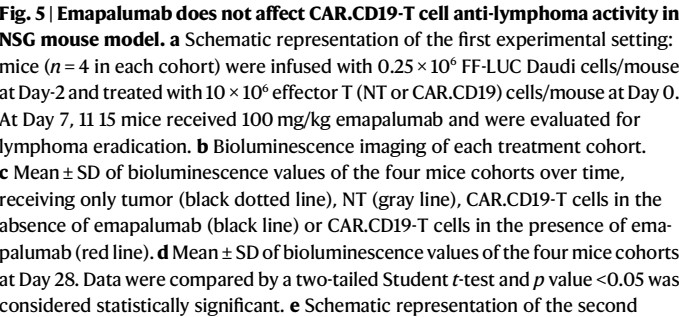

**Fig. 5 | Emapalumab does not affect CAR.CD19-T cell anti-lymphoma activity in NSG mouse model. a** Schematic representation of the first experimental setting: mice ($n = 4$ in each cohort) were infused with $0.25 \times 10^6$ FF-LUC Daudi cells/mouse at Day-2 and treated with $10 \times 10^6$ effector T (NT or CAR.CD19) cells/mouse at Day 0. At Day 7, 11 15 mice received 100 mg/kg emapalumab and were evaluated for lymphoma eradication. **b** Bioluminescence imaging of each treatment cohort. **c** Mean ± SD of bioluminescence values of the four mice cohorts over time, receiving only tumor (black dotted line), NT (gray line), CAR.CD19-T cells in the absence of emapalumab (black line) or CAR.CD19-T cells in the presence of ema-palumab (red line). **d** Mean ± SD of bioluminescence values of the four mice cohorts at Day 28. Data were compared by a two-tailed Student *t*-test and *p* value <0.05 was considered statistically significant. **e** Schematic representation of the second

experimental setting: mice ($n = 4$ in each cohort) were infused with $0.25 \times 10^6$ FF-LUC Daudi cells/mouse at Day-8 and treated with $10 \times 10^6$ effector T (NT or CAR.CD19) cells/mouse at Day 0. At Day 0, 3, 6 mice received 100 mg/kg emapa-lumab and evaluated for lymphoma eradication. **f** Bioluminescence imaging of each treatment cohort. **g** Bioluminescence values of each mouse, receiving NT cells in the absence of emapalumab (red line) or in the presence of emapalumab (red dotted line) or CAR.CD19-T cells in the absence of emapalumab (black line) or CAR.CD19-T cells in the presence of emapalumab (black dotted line). **h** Mean ± SD of bioluminescence values of the four mice cohorts at Day 15. Data were compared by a two-tailed Student *t*-test and *p* value <0.05 was considered statistically sig-nificant. Student's *t*-test was employed to calculate statistically significant differ-ences between groups. Source data are provided as a Source Data file.

Fig. 7, bottom panels) compared to mice not given emapalumab which develop severe CRS.

## Discussion

As CAR-T cell field is exponentially growing, scientists are seeking for novel strategies to mitigate, cure or even prevent the toxicities asso-ciated with treatment, therefore increasing the therapeutic window of the approach. However, the lack of comprehensive animal models able to recapitulate human toxicities represents a relevant obstacle. A humanized model of CRS was previously published[27], but the com-plexity of the approach employed has limited its large-scale applica-tion to further investigate CAR-T cell toxicity/pathophysiology and therapeutic interventions.

In order to study CAR-T cell-related complications, we developed a humanized mouse model that reproduces the most relevant human acute toxicities observed in patients after CAR-T cell infusion, namely CRS, ICANS and HLH/MAS. After humanization, mice were engrafted with a high tumor burden and received a standard dose of CAR-T cells. Thanks to the sub-lethal tumor cell irradiation, we were able to discern signs of CAR-T cell-derived toxicity, minimizing those related to lym-phoma engraftment and progression. The developed model is able to recapitulate severe CAR-T cell-related acute toxicities, and all mice died in the first 7 days following CAR-T cell infusion. Moreover, to the best of our knowledge, this is the first animal model in which a clear CAR-T cell-induced brain damage has been reported. In particular, after infusing a high number of tumor cells and a high dose of CAR-T cells, mice developed multifocal brain hemorrhages, which are well-

known manifestation of loss of cerebral vascular integrity with evi-dence of endothelial activation. Although brain hemorrhages are not formally included in the definition of ICANS, endothelial activation and multifocal vascular disruption are commonly found in the brain of patients developing fatal neurotoxicity after CAR-T cell infusion[12]. Indeed, it has been documented that patients with severe ICANS (grade ≥3) show evidence of endothelial activation, disseminated intravascular coagulation, capillary leak, and increased BBB perme-ability, leading to an increased risk of intracranial bleeding[12,28]. The permeable BBB fails to protect the cerebrospinal fluid (CSF) from the accumulation of systemic cytokines, including IFNγ, which induces brain vascular pericyte stress and secretion of endothelium-activating cytokines, ultimately promoting multifocal brain bleeding[29].

In our animal model recapitulating the CAR-T cell-related toxi-cities, we decided to investigate the role of INFγ neutralization. We focused our study on the use of emapalumab, because this humanized mAb, targeting both free and receptor-bound INFγ, is already clinically available and has shown a very promising toxicity profile in children with primary HLH relapsing, refractory or intolerant to conventional therapies, therefore representing an easily clinically translatable approach[23]. We documented the emapalumab ability to neutralize high concentration of IFNγ produced by CAR-T cells upon the engagement with tumor cells; we also showed that this neutralization does not affect the antitumor activity of CAR.CD19-T cells. This finding was first obtained in vitro, through the functional study of CAR.CD19-T cells cultured with lymphoma cells in the presence of high concentration of emapalumab, in standard co-culture conditions (E:T ratio of 1:1) and

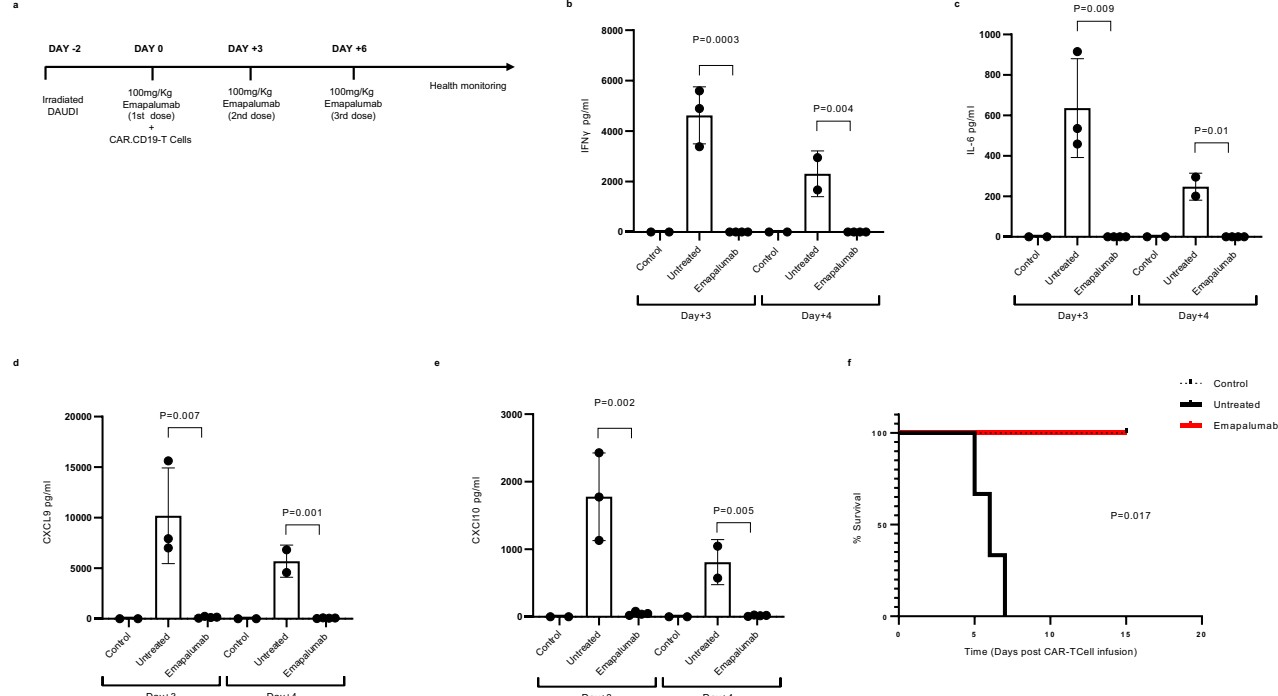

**Fig. 6 | Emapalumab displays therapeutic effect in humanized mice developing CRS. a** Schematic representation of the experimental design: hGM-CSF/hIL3 NOG mice were infused with irradiated FF-Luciferase positive sub-acutely irradiated Daudi cells (10 × 10⁶ cells/mouse). *n* = 10 total humanized mice) at Day-2 and treated with 10 × 10⁶ CAR.CD19-T cells/mouse in the presence of either 100 mg/kg/day emapalumab (*n* = 4) or vehicle (*n* = 4). Two mice were considered as for negative control infused only with DAUDI cells. **b** IFNγ, **c** IL-6, **d** CXCL9/MIG and **e** CXCL10/IP-10 levels were measured on peripheral blood of negative control (*n* = 2), treated

with 10×10⁶ CAR.CD19-T cells/mouse in the presence of either 100 mg/kg/day emapalumab (*n* = 4) or vehicle (*n* = 4) mice at Day+3 and Day+4 after effector T-cell infusion. Student's *t*-test was employed to calculate statistically significant differences between groups. Data are expressed as mean ± SD. All data above described were compared by a two-tailed Student *t*-test and *p* value <0.05 was considered statistically significant. **f** Kaplan–Meier survival curve analysis of CAR.CD19-T cell-treated mice without emapalumab (black line) or with emapalumab (red line). Source data are provided as a Source Data file.

then confirmed with a very low E:T ratio. Even in this unfavorable condition, the presence of emapalumab did not affect CAR-T cell activity. Proliferation assay and analysis of cytotoxicity at shorter timepoints confirm that CAR-T cell efficacy is preserved in the presence of emapalumab. It is noteworthy that these findings were obtained independently from the costimulatory domain included in the second-generation CAR construct targeting CD19, namely either 4.1BB or CD28, the latter being reported to be associated with the occurrence of more severe acute toxicities than those triggered by 4-1BB CAR.CD19-T cells[25].

Moreover, we have performed gene expression analysis to compare eight essential pathways related to CAR-T cell biology in either the presence or absence of IFNγ neutralization, showing that the only 14 genes (out of the 780 investigated) modulated in the presence of emapalumab were mainly genes associated with IFNγ signaling and toxicity pathways, as expected. This result is also corroborated by the flow-cytometry analysis of CAR.CD19-T cell with respect to the expression of activation markers, which reveals an identical activation profile between CAR.CD19-T cells exposed to emapalumab and the untreated counterpart. The preserved functionality of CAR-T cells in presence of emapalumab was confirmed also in vivo, in a lymphoma animal model, in the presence of high doses of emapalumab, administered with a schedule of 3 infusions, 4-day apart, concomitant with the onset of toxicity or concurrent with the infusion of CAR.CD19-T cell. No significant reduction in the activity of CAR.CD19-T cell against Daudi cells was observed in mice infused with emapalumab. In addition, in the NSG mouse model, our data indicate that CAR-T cell persistence and expansion were unaffected by IFNγ neutralization. Encouraged by these data, we then tested the possibility to control toxicity of CAR-T cells by using emapalumab in a comprehensive

humanized model. Notably, the administration of emapalumab induced a complete neutralization of IFNγ with a significant reduction of other inflammatory cytokines and chemokines, including CXCL9, a molecule shown to be highly correlated with HLH/MAS activity. In particular, we measured in mice blood, human IL-6, one of the most elevated cytokines during CRS, likely released by activated endothelial cells, as well as by activated macrophages stimulated by IFNγ, which represents the most relevant therapeutic target to manage CRS in humans. In patients, IL-6 elevation can cause capillary leakage, hypotension, activation of complement pathway and coagulation cascades, and myocardial dysfunction[30–32]. We clearly showed that the neutralization of IFNγ represents a valid approach to mitigate the elevation of this key component of the inflammatory cascade downstream of IFNγ itself. Indeed, we detected a significant reduction of circulating human IL-6 levels in mice treated with emapalumab, as compared to untreated mice. Moreover, in our animal model of CRS, we measured circulating levels of two human chemokines, CXCL9 (also known as MIG) and CXCL10 (also known as IP-10), specifically induced by IFNγ and synthesized and secreted by histiocytes and dendritic cells, thus allowing quantification of IFNγ-inhibition[33]. Both chemokines have been found to be significantly increased in patients experiencing grade 4–5 versus grade 0–3 CRS[34]. In our model, we found a significant increase of CXCL9 and CXCL10 in mice developing CRS after CAR.CD19 infusion. Notably, IFNγ neutralization was able to inhibit their elevation. Most importantly, emapalumab administration was able to completely protect mice from acute CRS, as all mice treated with emapalumab survived behind the timeline of the experimental plan (15 days), while untreated mice died within 7 days after CAR.CD19-T cell infusion. We also studied the brain damage in mice developing acute toxicity, and proved that emapalumab administration is able to

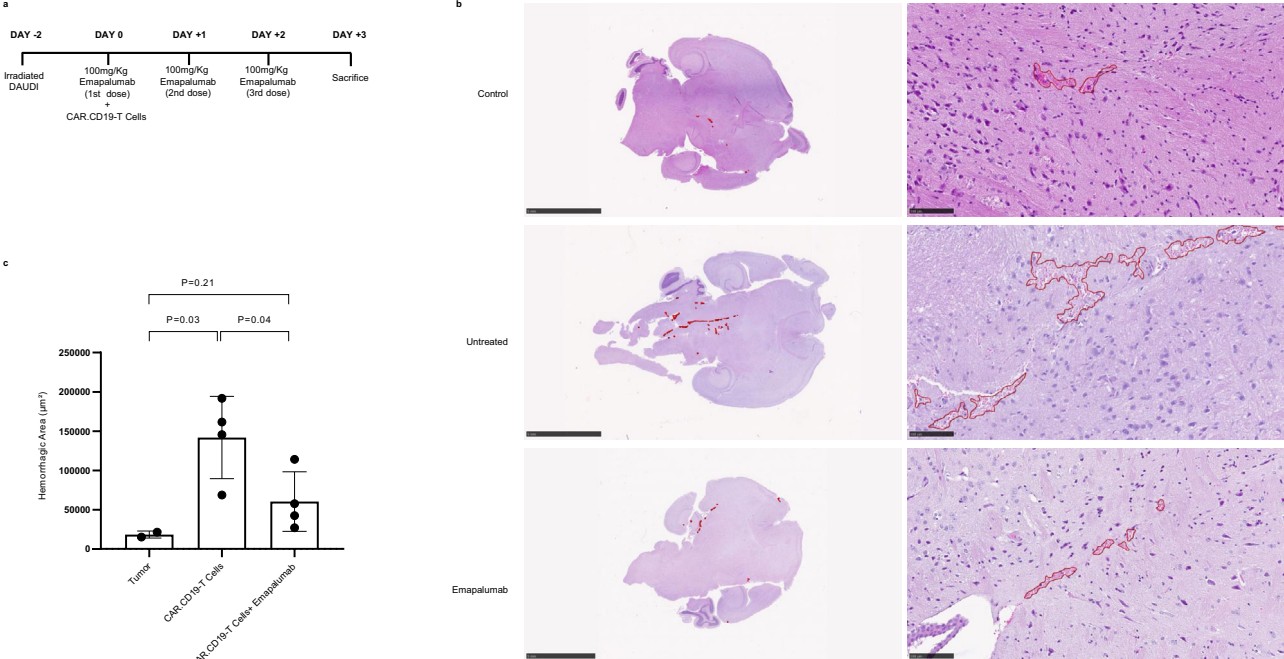

**Fig. 7 | Emapalumab prevents brain injury in humanized mice developing toxicity associated with CAR.CD19-T cells. a** Schematic representation of the experimental design: hGM-CSF/hIL3 NOG mice ($n = 10$) were infused with irradiated FF-Luciferase positive Daudi cells at Day-2 and treated with $10 \times 10^6$ CAR.CD19-T cells/mouse either in the presence of 100 mg/kg/day emapalumab ($n = 4$) or of the vehicle ($n = 4$), as reported in the cartoon. Two mice were considered for the negative control, being infused only with DAUDI cells (tumor). **b** Hematoxylin and eosin staining on brain slides of a control mouse (only tumor; top panels), untreated mouse (tumor and CAR.CD19-T cells in absence of emapalumab; middle panels) and treated mouse (tumor and CAR.CD19-T cells in presence of emapalumab; bottom panels). **c** Average of the hemorrhagic area in axial brain slide of control mouse (only tumor, $n = 2$), untreated mouse (tumor and CAR.CD19-T cells in the absence of emapalumab, $n = 4$) and treated mouse (tumor and CAR.CD19-T cells in the presence of emapalumab, $n = 4$). Data are expressed as mean ± SD. IHC staining has been performed on five slides from each murine brain. Data were compared by a two-tailed Student $t$-test and $p$ value <0.05 was considered statistically significant.

significantly control the multifocal bleeding in CNS, which has been associated with high-grade neurotoxicity in patients treated with CAR-T cells[12,28], as well as to downregulate the expression of key signaling molecules associated with IFNγ pathway and inflammation mediated by chemokines/cytokines. Altogether, these results provide the biological rationale for considering the use of emapalumab as pathogenetic treatment of ICANS.

Our data are in line with a recent experimental study demonstrating that an anti-INFγ approach, performed using a no azide/low endotoxin (NA/LE)-produced mouse anti-human IFNγ mAb, abrogates macrophage activation in an in vitro model of CRS[35]. However, in the study of Bailey et al.[35], the impact of IFNγ neutralization on severe systemic CRS, BBB damage or secondary HLH was not investigated in vivo. The potential importance of INFγ neutralization in management of CAR-T cell toxicities is supported by a recently published case report of a patient treated with emapalumab for life-threatening grade 4 CRS and neurotoxicity refractory to tocilizumab[36]. Emapalumab (administered at the dosage of 1 mg/kg) was infused on day 9 after numerous interventions, including repeated doses of tocilizumab, corticosteroids and siltuximab. Although it is not possible to attribute definitive causality to any particular intervention, patient fully recovered from CRS at day +18, and 1 week later from neurotoxicity. Notably, this patient maintained clinical remission and sustained B-cell aplasia for 12 months after CAR-T cell infusion, this observation corroborating our experimental observation that INFγ neutralization does not interfere with CAR-T cell efficacy.

In summary, in a simplified but comprehensive humanized model of CRS that overcomes the experimental limitations of previously reported CRS models, our study provides, for the first time, consistent evidence that emapalumab is a promising drug able to contain the CAR-T cell inflammatory cascade associated with high-grade CRS, MAS/HLH and brain injury, while sparing antitumor activity.

## Methods

### Research compliances

All mouse experimental procedures were approved by the Guidelines for Animal Care and Use of the National Institutes of Health (Ethical Committee for animal experimentation Prot. N 088/2016-PR). We have included the in vivo monitoring of the tumor growth by the use of IVIS Imaging system. As per the protocol, the maximum allowed bioluminescence was set to $10^{11}$ p/s/cm$^2$/sr.

The handling of human samples was conducted in accordance with the Institutional Review Board (IRB) of Bambino Gesù Children's Hospital, IRCCS, Rome, Italy (OPBG; Ethics Committee Approval N°969/2015 prot.N°669LB, and N°1422/2017 prot.N°810). Sex and gender information was not collected because it is not expected that sex and gender may affect CAR-T cell production or function.

### Cell cultures

CD19-positive human Burkitt's lymphoma cell lines Daudi (Cod. CCL-213, ATCC, USA) and Raji (Cod. CCL-86, ATCC, USA), genetically modified with firefly luciferase (FF-Luc), were cultured in RPMI 1640 medium (EuroClone, Italy) supplemented with 10% heat-inactivated fetal bovine serum (EuroClone), 2mM l-glutamine (GIBCO, USA), 25 IU/ml of penicillin, and 25 mg/ml of streptomycin (EuroClone), in a humidified atmosphere containing 5% CO$_2$ at 37 °C. All cell lines were authenticated by PCR-single-locus-technology (Promega, USA. PowerPlex 21 PCR) performed in "BMR Genomics s.r.l." (Italy), and routinely checked for mycoplasma (Venor®GeM Advance, MB Minerva Biolabs, UK) and surface marker expression.

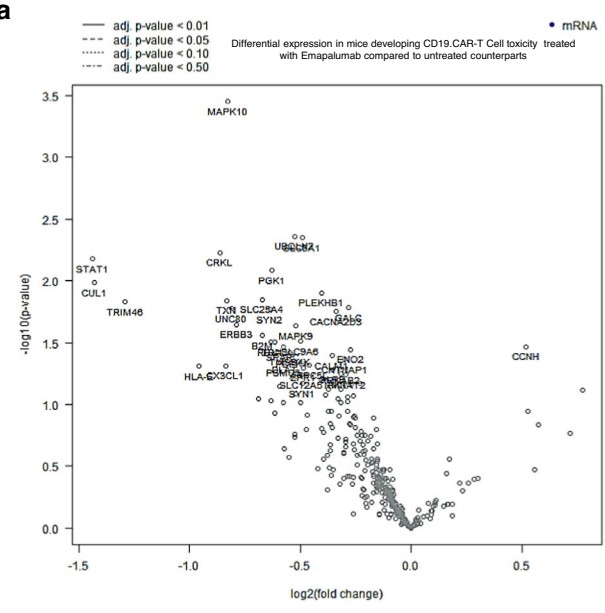

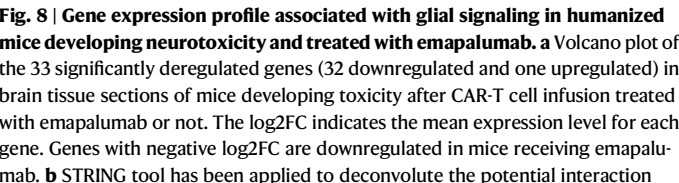

**Fig. 8 | Gene expression profile associated with glial signaling in humanized mice developing neurotoxicity and treated with emapalumab. a** Volcano plot of the 33 significantly deregulated genes (32 downregulated and one upregulated) in brain tissue sections of mice developing toxicity after CAR-T cell infusion treated with emapalumab or not. The log2FC indicates the mean expression level for each gene. Genes with negative log2FC are downregulated in mice receiving emapalumab. **b** STRING tool has been applied to deconvolute the potential interaction network among the 32 genes downregulated in brain tissue sections of mice treated with emapalumab, compared to brain tissues from control mice that did not receive the drug. **c** Gene enrichment analysis performed through the Enrichr web tool displays the pathways in which the 32 genes downregulated in emapalumab-treated mice are involved. For differential expression analysis, a *p* value of ≤0.05 (calculated with the nSolver™ 4.0 analysis software), was applied as cutoff. Source data are provided as a Source Data file.

## CAR.CD19-T cells generation
PBMC derived from buffy coats of HDs were isolated by ficoll gradient separation, activated by exposure to immobilized OKT3 (1 µg/ml, e-Bioscience Inc.; San Diego, CA, USA) and anti-CD28 (1 µg/ml, BD Biosciences, Europe) mAb in the presence of recombinant human interleukin-7 (IL7, 10 ng/ml; R&D; USA) and 15 (IL15, 5 ng/ml; R&D), transduced by retroviral vector above described and expanded[37].

The retroviral CAR.CD19 construct includes the anti-human CD19-scFv derived from FMC63 clone, in frame with 16aa sequence of the human CD34 antigen, the CD8 stalk domain, the CD8 transmembrane domain and the 4.1BB and CD3ζ cytoplasmic domains[37]. The retroviral CD28-based CAR.CD19 construct includes the anti-human CD19-scFv derived from FMC63 clone, in frame with 16aa sequence of the human CD34 antigen, the CD8 stalk domain, the CD8 transmembrane domain and the CD28 and CD3ζ cytoplasmic domains. Viral supernatant was generated by the use of the producer cell line 293VEC-RD114 (kindly provided by BioVec Pharma, Canada).

## Phenotypic analysis
Expression of cell surface molecules was determined by flow-cytometry using standard methodology. The mAbs used in this study were listed in Supplementary Table 3. Cells were collected from co-cultures, washed with PBS and incubated with the antibodies as indicated by the manufacturer. Samples were acquired with a BD LSRFortessa X-20 and analyzed using the FACSDiva software (BD Biosciences). For each sample, we analyzed a minimum of 20,000 events.

## Cytokine analysis
IFNγ, Granzyme B, IL-2, TNF-α, IL-6, IL-1β, CXCL9/MIG and CXCL10/IP-10 levels found either after 24 h co-culture assays or in peripheral blood of mice were analyzed in the enzyme-linked lectin Ella Automated Immunoassay System Instrument (Bio-Techne, California, USA), by Simple Plex software.

## Cytotoxicity assays
The IncuCyte S5 Live Cell Assay System (Sartorius, Michigan, MI, USA) was used for kinetic monitoring of CAR-T cell cytotoxicity against tumor in the presence or absence of emapalumab. Effector T cells (NT or CAR.CD19) and FF-LUC Daudi tumor cells were plated at $0.25 \times 10^6$ cells/well on 48-well plates at 1:1 E:T ratio in the presence or absence of 100 µg/ml emapalumab and incubated for 64 h at 37 °C in a humidified atmosphere (5% $CO_2$). Growth curves were generated by the algorithm in the "2021C" software (Schrödinger; New York, NY, USA) evaluating GFP values expressed by Daudi tumor cells from data points acquired during imaging at 2-h intervals. All samples were plated in triplicate.

Short-term (6 h) CAR.CD19-T cell cytotoxic assay was performed with a luciferase-based test. Briefly, effector T cells (NT or CAR.CD19) and FF-LUC-expressing Raji tumor cells were plated, at $0.5 \times 10^6$ cells/well, on 24-well plates, at 1:1 E:T ratio, either in the presence or in the absence of 100 µg/ml emapalumab, and incubated at 37 °C in a humidified atmosphere (5% $CO_2$). After 6 h, 15 µg/ml of luciferin (Xenolight D-luciferin; Perkin Elmer) was added to the media and bioluminescence (BLI) was quantified after 10 min of incubation at 37 °C in a humidified atmosphere (5% $CO_2$) on an ENSPIRE Multimode plate reader (Perkin Elmer) by Inspire Manager software.

## Apoptosis assay
Apoptosis of Raji target cells was evaluated by flow-cytometry-based assay. In particular, effector T cells (NT or CAR.CD19) and Raji tumor cells were plated at $0.5 \times 10^6$ cells/well, on 24-well plates, at 1:1 E:T ratio,

either in the presence or in the absence of 100 µg/ml emapalumab, and incubated at 37 °C in a humidified atmosphere (5% $CO_2$). After 6 h, cells were labeled with Annexin V BUV-395 (BD Biosciences)/ 7-Amino-Actinomycin D (7-AAD; BD Biosciences) according to the manufacturer's instructions and evaluated by flow-cytometry, as specified in the phenotypic analysis.

### Cell proliferation assay
Cell proliferation assay was performed by labeling cells with the Cell-TraceTM Cell Proliferation Kit (Far Red, Thermo Fisher Scientific) according to the manufacturer's instructions. In brief, CAR.CD19-T cells were washed with PBS and resuspended at $10^6$ cells/ml in a working dye solution (1 µM in 1xPBS) for 20 min at 37 °C. Thereafter, labeled cells were resuspended in five volumes of cell culture medium, centrifuged, and resuspended in culture media to be co-cultured with Daudi tumor cells (at 1:1 E:T ratio) in the presence or absence of 100 µg/ml emapalumab for up to 3 days.

### Nanostring analysis
RNA was isolated from samples using the RNeasy mini kit (Qiagen) according to the supplier's instructions. Total RNA was quantified with NanoDrop ND-100. We analyzed the expression of 780 genes (including 10 reference genes) related to components of CAR-T biology using the nCounter CAR-T Characterization Panel™ (XT CAR-T Code-Set Panel, Nanostring, Seattle, WA) and of 770 genes (including 10 reference genes) related to components of glial biology using the nCounter Human Glial Profiling panel (XT Human Glial Profiling Code-Set Panel, Nanostring, Seattle, WA). Total RNA was used as input and sample hybridization was performed according to the manufacturer's instructions. Sample detection and analysis were completed on an nCounter® Digital Analyzer. Raw data processing, quality control, and normalization were performed using the nSolver™ 4.0 analysis software (NanoString nCounter Technologies, Seattle, WA). Background subtraction from raw transcript counts was performed through negative input controls. Normalization to 10 housekeeping genes and differential expression analysis were completed using the Advanced Analysis software plugin (version 2.0.115). For differential expression analysis, a p value of ≤0.05, was applied as a cutoff. Gene set enrichment was evaluated with the Enrichr (https://maayanlab.cloud/Enrichr/) web tool. The basic interaction unit in STRING has been used to evaluate possible network between genes differentially expressed (https://string-db.org/).

### In vivo CAR+ lymphoma mouse model
To perform antitumor in vivo study, Cg-Prkdcscid Il2rgtm1Wjl/SzJ (NSG) 6-week-old female mice were purchased from Charles Rives Laboratories and maintained in the Plaisant animal facility in Castel Romano, Rome, Italy. Mice were i.v. engrafted with $0.25 \times 10^6$ FF-Luciferase positive Daudi cells at Day-2 or at Day-8, and treated with $10 \times 10^6$ non-transduced (NT) or CAR.CD19-T cells/mouse at Day 0. At Day 7, 11, 15 (Fig. 5a) or at Day 0, 3, 6 (Fig. 5d), respectively, mice received 100 mg/kg Emapalumab. Tumor growth was monitored weekly by IVIS Imaging System[38] and IVIS living image software after D-Luciferin (Perkin Elmer, D-Luciferin potassium salt) i.p. administration.

### Humanized murine model of CRS and brain injury after CAR.CD19-T cell administration
Fourteen-week-old female CIEA NOG-EXL (hGM-CSF/hIL3 NOG) mice were purchased from Taconic Laboratories and maintained at the Plaisant animal facility in Castel Romano, Rome, Italy. Mice were engrafted with human umbilical cord blood-derived CD34+ HSCs. Mice were aged 10 weeks post engraftment and quality checked for human leukocyte reconstitution by flow-cytometry. Only mice reaching ≥25% hCD45+ cells were used in the experiments. Humanized mice were infused with irradiated (30 cGy) FF-Luciferase positive Daudi cells at Day-2 and treated either with $10 \times 10^6$ NT-T cells or with CAR.CD19-T cells/mouse either in the presence or absence of 100 mg/kg/day emapalumab, according to the treatment schedules. Mice were subject to blood bleeding at Day 3 and Day 4 for the monitoring of cytokines. For brain injury evaluation, 3 days after CAR.CD19-T cell infusion, animals were sacrificed, and brains were fixed in 4% formaldehyde in 0.1 M phosphate buffer (pH 7.2) and paraffin embedded. The histopathologic hematoxylin and eosin staining was performed on the middle axial sections[39]. For HLH/MAS bone marrow analysis, tibiae of experimental mice were collected at sacrifice, and tissues were fixed with 10% neutral buffered formalin and embedded in paraffin. Deparaffined sections were stained with hematoxylin and eosin (Thermo Fischer Scientific). Images were acquired on a ScanScope XT scanner and digitized by NDPVIEW software.

### Statistics and reproducibility
Unless otherwise noted, data are summarized as mean ± standard deviation (SD). Student's t-test or Mann–Whitney test (two-sided) were used to determine statistically significant differences between samples, with a p value <0.05 indicating a significant difference.

Functional data performed in in vitro experiments are representative of at least three experiments with different human donors. The data are displayed as scatter dot plots and individual points represent replicates of different HDs.

Mouse survival data were analyzed using Kaplan–Meier survival curves and the log-rank test was used to measure differences between groups. No valuable samples were excluded from the analyses. Mice were randomized based on the tumor signal for control and treatment groups before infusion of NT or CAR-T cells. To compare the growth of tumors over time, bioluminescence signal intensity was collected blindly. Bioluminescence signal intensity was log-transformed and, then, compared using a two-sample t-test. We estimated the sample size considering no significant variation within each group of data. The principle of using the smallest sample size possible was adopted in planning the animal experiments. We estimated the sample size in order to detect a difference in averages of 2 SDs at the 0.05 level of significance with an 80% power. The data are displayed as scatter dot plots and individual points represent replicates of each different experimental cohort.

Graphic representations and statistical analysis were performed using GraphPad Prism 6 (GraphPad Software, La Jolla, CA). IHC staining has been performed on five slides from each investigated tissue.

The raw data are available in the Source Data file.

### Reporting summary
Further information on research design is available in the Nature Portfolio Reporting Summary linked to this article.

## Data availability
The Nanostring data generated in this study have been deposited in the GEO database under accession code GSE228924 (https://www.ncbi.nlm.nih.gov/geo/query/acc.cgi?acc=GSE228924). Source Data are provided with this paper.

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

## Acknowledgements

The experimental work was supported by grants awarded by Accelerator Award – Cancer Research UK/AIRC – INCAR project (F.L.), Associazione Italiana Ricerca per la Ricerca sul Cancro (AIRC)-Special Project 5×1000 no. 9962 (F.L.), AIRC IG 2018 id. 21724 (F.L.), MFAG 21979 (C.Q.), id.26915-2021 AIRC Fellowships Call (S.M.), Ricerca Corrente (C.Q., B.D.A.), Ministero dell'Università e della Ricerca (Grant PRIN 2017 to F.L.); Italian Healthy Ministry project on CAR-T RCR-2019-23669115 (Coordinator F.L.), GR-2016-02364546 (B.D.A.), RF-2016-02364388 (F.L.), RF-2021-12374120 (C.Q.), Independent Research grant AIFA (F.L.: 2016 call), "PNRR M4C2- Investment 1.4-CN00000041"– NextGenerationEU, PNC HLS-TA 2022 (F.L.). We are very grateful to SOBI for the collaboration in providing emapalumab used for the in vitro and in vivo

experiments. We are very grateful to BioVec Pharma and Dott. Manuel Caruso in providing the packaging 293Vec cell lines.

## Author contributions

C.Q., B.D.A., and F.L. designed experimental studies, supervised the project conduction, analyzed the data and wrote the manuscript. S.M., F.D.B. and P.M. equally contributed to the manuscript. S.M., D.A.S., M.G., S.C., S.R., L.I., M.P., S.D.C., M.S., A.O., M.C.Q., M.A., A.S., R.C., Z.A., and M.C. developed the in vitro models and performed the in vitro experiments. S.M., F.D.B., M.C., and B.D.A. performed the in vivo experiments. S.R. and D.A.S. performed bioinformatics analysis on NanoString data. C.Q. and B.D.A. cloned the retroviral vector. S.M. and M.S. performed FACS analysis. R.D.V. performed HLA/MAS evaluation in the in vivo model. F.D.B., P.M., A.M., M.C.L., M.G.C., and F.L. provided healthy donor material, medical advices and expertise in CAR-T cell toxicity, as well as in the use of emapalumab.

## Competing interests

F.L. and P.M. declare the following competing interests: on September 2022 participated in an advisory board on primary HLH organized by Sobi, receiving honoraria. All the other authors do not have any competing interests to disclose. SOBI has not supported the study, besides the provision of the emapalumab drug.
