## [Peer Review File · Nature Communications]

Neutralizing IFN γ improves safety without compromising efficacy of CAR-T cell therapy in B-cell malignanciesReviewers' comments:

Reviewer #1 (Remarks to the Author):

In this manuscript by Manni et al, the authors use in vitro and in vivo models to confirm recent reports that IFN γ is not required for CAR-T efficacy and that targeting this cytokine reduces CAR-T-mediated toxicity. The major findings of this manuscript is a more scalable and humanized mouse model for potential CRS studies, and the use of this model to test the IFN γ effects on both CRS and ICANS in vivo is a potentially significant advancement to the field. However, the depth of the data-driven description of the model is still lacking, and there are insufficient comparators and controls in the current presentation. I have several concerns with the data presented.

Major concerns:

1. Given the importance of the in vivo work proposed herein, more details and data regarding the model itself should be included. Please show the engraftment data for CD45 that is mentioned in the text. What cell subsets engraft (B cells? T cells? Macrophages? Neutrophils?) Given the possible engraftment of T cells, how does this affect the injection of allogeneic CAR-T? What is the frequency of graft loss after injection of allogeneic T cells? What about rejection of the incoming CAR T cells? (on what time scale and dose does this occur)?

2. Given the short-lived nature of the model, tumor burden shouldn't be an issue in assessing toxicities. Have you tried using non-irradiated tumor cells in your humanized model instead of irradiated?

3. The dosing schedule for emapalumab is different for every mouse experiment in the paper, which raises concerns. In the initial experiment, it was administered weekly starting at day 7. Did this provide complete coverage? Was IFN γ in the serum tested to confirm blockade? Although this mimics the clinical schedule, CAR-T-mediated cytokines peak earlier in mice so it's likely that adding it at day 7 is past the peak of IFN γ expression. In the second experiment, mice were given emapalumab at days 0, 3 and 5 and then the final experiment was 0, 1 and 2. If injection once a week provided full coverage, doses every day or 2-3 days shouldn't be necessary and raises unnecessary variables in the paper.

4. While IL-6, CXCL9 and CXCL10 play a role in CRS/neurotoxicity, there are a host of additional cytokines/chemokines (i.e. TNF α , MIP1 α , IL-10, MCP1, etc.) that could be measured in blood as indicators of T cell activity. Did the authors measure any of these?

5. The findings of brain hemorrhages are intriguing as a model of ICANS. Can the authors also include a tocilizumab-treated control? It would be interested to see if tocilizumab treatment made the brain hemorrhages better (which is different than human cases), or the same or worse (which would approximate human cases).

6. The authors tested only CARs with a 4-1BB signaling domain. Can the authors expand their studies to include CD28-signaling CARs?

7. What about impact on CAR T cell persistence? Did the authors measure CAR T cell expansion or engraftment in their models? Given that these are allogeneic T cells, at what timepoint and doses do these mice develop GvHD or rejection of the graft?

Minor concerns:

1. What are the stats referring to in Fig. 1 E/F? Which groups are being compared?

2. Interferon gamma is sometimes referred to as IFN γ and other times INF γ . Please make this consistent.

3. line 305: what is NA/LE (in reference to mouse anti-human IFN γ monoclonal antibody)?

Reviewer #2 (Remarks to the Author):

Manni and colleagues present data to suggest that complications seen with CAR T cell therapies is related at least in part to the secretion of IFN γ . The rationale for this hypothesis, at least with respect to neurologic toxicity (ICANS), is reasonably sound given the impact of IFN γ on the BBB. The authors initially demonstrate that CAR T cells retain cytotoxic function in the setting of IFN γ neutralizing antibodies and a resulting change in the T cell gene expression which was limited to only 14 genes. The combination of CD19 targeted CAR T cells with IFN γ blockade had no impact on anti-tumor efficacy in a xenograft tumor model. Utilizing a humanized mouse model with irradiated tumor cells, the investigators demonstrated safety given the fact that in this model mice developed lethal CRS and brain hemorrhages when treated with CAR T cells in the absence of IFN γ blocking MAbs. This is a well written manuscript. There are some additional questions which need to be addressed.

1. In the in vitro studies presented, the authors demonstrate no impact of IFN γ blockade on CAR T cell cytotoxic activity which may not be surprising. However, T cell expansion is a critical feature of CAR T cells which expand exponentially in vivo (as well as in vitro with resimulations). The authors need to include data to support that proliferative potential is maintained in the absence of IFN γ stimulation. This is critical.
2. The NSG in vivo studies use a very high dose of CAR T cells wherein T cell in vivo expansion and IFN γ secretion may not be relevant. These studies should be done with lower "stress doses" wherein CAR T cell proliferation is needed for tumor eradication to confirm the authors' conclusions (i.e. down to 100,000 cells per mice).
3. Along the same lines, it would be relevant that in the context of IFN γ blockade, that in vivo T cell proliferation actually occurs.
4. I'm not sure whether irradiated tumors are appropriate for the humanized mouse model wherein the antigen load would be more relevant. CRS is seen most commonly in the context of high tumor burden. This is not the case with the presented CRS/ICANS mouse model presented. What happens in the context of high tumor burden (more clinically relevant), when mice are treated with CAR T cells? Does IFN γ blockade still prevent CRS?
5. Brain hemorrhages seen is not classical for ICANS, are there other markers for ICANS which can be demonstrated in this model to be more compelling? Have the authors looked at other markers for HLH such as coagulopathy, ferritin etc?

Reviewer #3 (Remarks to the Author):

The manuscript submitted by Manni et al. aims to address the mechanisms underlying CRS and neurotoxicity common side effects observed in patients with B-cell malignancies after CAR-T cell infusion. The authors use in vitro and in vivo models to assess the role of IFN-gamma in these side effects by using an FDA-approved blocking monoclonal antibody, emapalumab. Even if this is a highly relevant topic in the field and the concept may be of interest, I consider that the study is quite underdeveloped at this stage. More robust experimental data should be provided to conclude that blocking IFN-gamma decreases CAR T-cell-induced side effects while maintaining unaffected anti-cancer efficacy. Also, the study at its present form has some important deficiencies.

First, a deeper characterization of the effects of IFN-gamma blocking on both CAR-T and target cells is missing. In Fig.1 it is not clear to me why authors measure target cell killing at 7 days of co-culture, what does happen at shorter timepoints (e.g. 16 or 24 h)? Kinetics of cell death, state of apoptotic mediators, etc. should be evaluated. Fig.2 aims to characterize the effects of IFN-gamma blocking in CAR-T cells, however these results could not be evaluated due to poor quality/resolution. In addition, a broader characterization of T cell activation should be evaluated at the protein level, not only measuring CD25 and CD69 activation markers (Suppl Fig. 2). It is not clear to me why CAR-T cell activation was measured with Fc-CD19 instead of using target lymphoma cells, which would provide more valuable information. In Fig. 3 anti-cancer efficacy is determined by injecting CAR-T cells two days after inoculating lymphoma cells, while first signs of detectable tumor burden are observed at day 14. A therapeutic model with detectable tumor burden should be used to resemble the situation in patients. Moreover, emapalumab is

administrated 7 days after CAR-T injection, which does not seem to be the best setting to test if IFN-gamma blocking does not affect anti-cancer efficacy. Emapalumab should be rather administrated concomitantly with CAR-T cells, similarly to the following settings aiming to assess CAR-T side effects (Figs. 4-6). The study also lacks a characterization of CAR-T cells in vivo after IFN-gamma blocking. Is this affecting CAR-T cell expansion? Finally, the evidence supporting that IFN-gamma blocking reduces CAR-T cell side effects is not convincing. In particular, using only hematoxylin/eosin to assess multifocal haemorrhagic lesions and/or macrophage activating syndrome in bone marrow seems limited. In addition, the manuscript does not clearly state how many animals are being analyzed in each experiment and how many times each experiment was repeated. Also, it is not clear which statistics analyses were performed in each case.

I would recommend addressing the above issues prior to consider submitting this manuscript for publication.

Dear Reviewers,

thank you very much for all the suggested revisions aimed at improving the quality of our paper.

We have significantly modified the original version of the manuscript, including the recommended experiments to address all the concerns raised. We implemented the pre-existing Figures with new data and added more information to support our observations.

We hope to have addressed all issues to allow you to reconsider this manuscript suitable for publication.

Reviewers' comments:

Reviewer #1 (Remarks to the Author):

In this manuscript by Manni et al, the authors use *in vitro* and *in vivo* models to confirm recent reports that IFN γ is not required for CAR-T efficacy and that targeting this cytokine reduces CAR-T-mediated toxicity. The major findings of this manuscript is a more scalable and humanized mouse model for potential CRS studies, and the use of this model to test the IFN γ effects on both CRS and ICANS *in vivo* is a potentially significant advancement to the field. However, the depth of the data-driven description of the model is still lacking, and there are insufficient comparators and controls in the current presentation. I have several concerns with the data presented.

Major concerns:

1. Given the importance of the *in vivo* work proposed herein, more details and data regarding the model itself should be included. Please show the engraftment data for CD45 that is mentioned in the text. What cell subsets engraft (B cells? T cells? Macrophages? Neutrophils?) Given the possible engraftment of T cells, how does this affect the injection of allogeneic CAR-T? What is the frequency of graft loss after injection of allogeneic T cells? What about rejection of the incoming CAR T cells? (on what time scale and dose does this occur)?

We thank the Reviewer for recommending to better clarify these aspects and to more comprehensively describe our *in vivo* humanized model. We have now included a novel Figure (Supplemental Figure 5) to provide more data regarding the engraftment of human CD45+ cells and the relative cell subset distribution in the humanized model. Moreover, we have included the analysis of CAR.CD19 T-cell detection at the last day (Day +3) of the experimental setting, proving that CAR

T-cells were not promptly rejected by the allogenic T-cells engrafted in the mice. We would like to comment on the fact that the model we developed is for severe and life-threatening CRS, and thus mice were sacrificed very early after CAR T-cell infusion. The included data support the hypothesis that the rejection of CAR T-cells does not occur in the timeframe of the experiment.

2. Given the short-lived nature of the model, tumor burden shouldn't be an issue in assessing toxicities. Have you tried using non-irradiated tumor cells in your humanized model instead of irradiated?

We thank the Reviewer for raising this issue which offers us the opportunity to better elaborate on this point. The development of CRS requires the use of high tumor burden to achieve CAR T-cell activation level, leading to the onset of toxicity symptoms mimicking those typical of the CRS observed in patients. We have performed the first pilot experiment to define our CRS model by infusing non-irradiated tumor cells and letting them grow in the humanized mice before CAR T-cell infusion to reach high tumor burden. Unfortunately, using this approach, it was not possible to discriminate between the CRS signs and those related to tumor overgrowth. Indeed, the overall survival of the mice receiving only non-irradiated Daudi and those in which we have infused also CAR T-cells was superimposable. For this reason, to reproduce an *in vivo* model that recapitulates human CRS, we infused a high tumor burden of irradiated tumor cells, capable of activating CAR.CD19 T-cells, but not generating confounding signs of toxicity in the mice.

3. The dosing schedule for emapalumab is different for every mouse experiment in the paper, which raises concerns. In the initial experiment, it was administered weekly starting at day 7. Did this provide complete coverage? Was IFN γ in the serum tested to confirm blockade? Although this mimics the clinical schedule, CAR-T-mediated cytokines peak earlier in mice so it's likely that adding it at day 7 is past the peak of IFN γ expression. In the second experiment, mice were given emapalumab at days 0, 3 and 5 and then the final experiment was 0, 1 and 2. If injection once a week provided full coverage, doses every day or 2-3 days shouldn't be necessary and raises unnecessary variables in the paper.

We thank the Reviewer for these valuable comments. We have now better clarified the rationale behind the chosen emapalumab schedule, and we have added more *in vivo* experiments to reduce the variability in the emapalumab administration. The initial experimental design (Figure 5A) was performed to investigate the anti-tumor activity of CAR.CD19 T-cells in the setting of a limited tumor burden (as in the majority of the cases in which CAR T-cells are applied in the real-world). In our

clinical experience, in this patient setting, CAR T-cells will need around 7 days to start the *in vivo* expansion, a time frame in which we also start to observe fever and CRS. Thus, in our murine model, we have shown that on Day 7 after CAR T-cell infusion, mice were characterized still by a significant bioluminescence, this suggesting that the lymphoma was not yet eradicated. For this reason, we have decided to start the infusion of emapalumab on day 7.

As per the Reviewer suggestion, we have now included a second *in vivo* animal setting, in which emapalumab has been used with a preventive approach, by infusing the drug in the same day of CAR T-cell infusion (Figure 5E). In this case, we have challenged the model by recapitulating the clinical setting in which CAR T-cells are infused in patients with high tumor burden. The rationale behind this emapalumab schedule is that in patients with high tumor burden we often observe a more severe CRS occurring few hours after the CAR T-cell infusion, and likely these patients are the ones who could benefit from a preventive administration of emapalumab. In our experimental setting, we have provided evidence that the IFN γ neutralization occurring at the same day of the CAR T-cell infusion did not affect the anti-lymphoma activity. Notably, with this schedule of infusion at day 0, we have recapitulated the same approach also used to control CRS occurrence in the humanized animal model (Figure 6A), as requested by the Reviewer.

We have also included the measurement of IFN γ levels during the performed *in vivo* study (Supplemental Figure 3A), showing that IFN γ is neutralized by emapalumab up to the end of the study. We administered emapalumab every 3 days to recapitulate the schedule of the drug administration used in the clinical setting (N Engl J Med 2020; 382:1811-1822).

4. While IL-6, CXCL9 and CXCL10 play a role in CRS/neurotoxicity, there are a host of additional cytokines/chemokines (i.e. TNF α , MIP1a, IL-10, MCP1, etc.) that could be measured in blood as indicators of T cell activity. Did the authors measure any of these?

We concur with the Reviewer that a comprehensive cytokine analysis should also include other soluble factors, and, thus, we have now added data on IL-1 β (Supplemental Figure 6A) and TNF α (Supplemental Figure 6B) cytokines. For these cytokines a clear trend of modulation is observed, although the difference with the level of cytokine detected in mice not treated with emapalumab was not statistically significant. Thanks a lot for this helpful suggestion.

5. The findings of brain hemorrhages are intriguing as a model of ICANS. Can the authors also include a tocilizumab-treated control? It would be interested to see if tocilizumab treatment made the brain hemorrhages better (which is different than human cases), or the same or worse (which

would approximate human cases).

We agree with the Reviewer that it is a relevant question; however, we also think that the Reviewer will concur with us that this issue is outside the scope of our work. The CRS model that we developed will be further applied in future studies aimed at investigating the role of tocilizumab, as well as of other agents in the control of CAR T-cell-associated neurotoxicity.

6. The authors tested only CARs with a 4-1BB signaling domain. Can the authors expand their studies to include CD28-signaling CARs?

We thank the Reviewer for this helpful suggestion. In the new version of the manuscript, we have included *in vitro* cytotoxicity data with CAR.CD19 T-cells including the CD28 costimulatory domain, *ad hoc* cloned in our unit to address the request of the Reviewer. The data we obtained are included in the new Figure 2. Also in this context, the presence of emapalumab (100ug/ml) does not alter the behavior of the CD28-based CAR.CD19 T-cells neither in terms of long-term cytotoxicity (Figure 2A), nor of short-term cytotoxicity (Figure 2C), nor of proliferation activity (Figure 2D). Furthermore, complete neutralization of IFN γ does not result in modulations of Granzyme B, IL2, TNF α (Figure 2B). Thanks a lot for this helpful suggestion.

7. What about impact on CAR T cell persistence? Did the authors measure CAR T cell expansion or engraftment in their models? Given that these are allogeneic T cells, at what timepoint and doses do these mice develop GvHD or rejection of the graft?

We thank the Reviewer to underline this aspect. In the new version of the manuscript, we have reported the data concerning the persistence and expansion of CAR T-cells both in the NSG model used to evaluate the antitumor activity in the presence of emapalumab and in the humanized NSG model employed to study CRS. In the NSG model used for antitumor evaluation, CAR T-cell expansion was observed in the mouse peripheral blood approximately one month after infusion, and as shown in Supplementary Figure 3B-D, CAR T-cell persistence and expansion were unaffected by IFN γ neutralization (the serum level of IFN γ is reported in Supplementary Figure 3A). However, as suggested by the Reviewer, since CAR T-cells are allogeneic cells, mice developed xenograft two months after the infusion of CAR T-cells. Thus, CAR T-cells were not rejected in this model (Supplemental Figure 3).

In the context of the humanized model, we underline that the setting of acute toxicity imposed us to early sacrifice the mice. In this short time frame, neither xenoreaction, nor CAR T-cell rejection were observed (Supplemental Figure 7).

Minor concerns:

1. What are the stats referring to in Fig. 1 E/F? Which groups are being compared?

The heatmap graph does not allow to include the stats, but we have reported in the text and the legend of the figure that we have observed a significant modulation in the IFN γ detection between treated and untreated conditions (p-value calculated by t-student test and was lower than 0.05 in all the tested condition), whereas no difference was observed for the other cytokines that we investigated.

2. Interferon gamma is sometimes referred to as IFN γ and other times INFg. Please make this consistent.

We apologize for this inconsistency; we have now referred to IFN γ across the whole manuscript.

3. line 305: what is NA/LE (in reference to mouse anti-human IFN γ monoclonal antibody)?

We apologize to have missed the acronym NA/LE description in the text. We have now specified it: with no azide and low endotoxin (NA/LE) Ab. In particular, the NA/LE IFN γ -Ab has been used by Bailey, S. R. et al. to perform in vitro culture or in vivo experiments (for non-human studies), as well as for functional assays — blocking, neutralizing, activation or depletion — where the presence of azide may damage cells or exogenous endotoxin may signal or activate cells.

Reviewer #2 (Remarks to the Author):

Manni and colleagues present data to suggest that complications seen with CAR T cell therapies is related at least in part to the secretion of IFN γ . The rationale for this hypothesis, at least with respect to neurologic toxicity (ICANS), is reasonably sound given the impact of IFN γ on the BBB. The authors initially demonstrate that CAR T cells retain cytotoxic function in the setting of IFN γ neutralizing antibodies and a resulting change in the T cell gene expression which was limited to only 14 genes. The combination of CD19 targeted CAR T cells with IFN γ blockade had no impact on anti-tumor efficacy in a xenograft tumor model. Utilizing a humanized mouse model with irradiated tumor cells, the investigators demonstrated safety given the fact that in this model mice developed lethal CRS and brain hemorrhages when treated with CAR T cells in the absence of IFN γ blocking MAbs. This is a well written manuscript. There are some additional questions which need to be addressed.

1. In the *in vitro* studies presented, the authors demonstrate no impact of IFN γ blockade on CAR T cell cytotoxic activity which may not be surprising. However, T cell expansion is a critical feature of CAR T cells which expand exponentially *in vivo* (as well as *in vitro* with re-stimulations). The authors need to include data to support that proliferative potential is maintained in the absence of IFN γ stimulation. This is critical.

We thank the Reviewer for this important observation. We agree that it is critical to include data showing that the proliferative ability of CAR T-cells is maintained in the presence of IFN γ neutralization. For this reason, in the new version of the manuscript, we have included the results concerning the expansion of CAR T-cells either in the presence or in the absence of IFN γ neutralization obtained both *in vitro* and *in vivo* NSG model. In the *in vitro* experiments, we have proved that both CAR.CD19.4-1BB and CAR.CD19.CD28 T cells stimulated through the coculture with CD19+ lymphoma cells have comparable expansion in the presence or absence of saturating concentration of emapalumab (Figure 1E and Figure 2D).

Moreover, we have been also able to corroborate this *in vitro* data with the *in vivo* animal model. As shown in Supplementary Figure 3B-D, in the NSG model, CAR T-cell expansion was unaffected by IFN γ neutralization (IFN γ level in the serum of treated mice has been reported in Supplementary Figure 3A). Thanks a lot for this helpful suggestion.

2. The NSG *in vivo* studies use a very high dose of CAR T cells wherein T cell *in vivo* expansion and IFN γ secretion may not be relevant. These studies should be done with lower “stress doses” wherein CAR T cell proliferation is needed for tumor eradication to confirm the authors’ conclusions (i.e. down to 100,000 cells per mice).

We thank the Reviewer for this helpful suggestion. Our experiments were carried out in the optimized and standard conditions (with 10×10^6 CAR T-cells/mouse) for the eradication of the tumor. However, following the Reviewer's suggestion, in the new version of the manuscript, we have included data from a new *in vivo* experiment performed under suboptimal conditions, using also lower “stress doses”, namely 1×10^6 and 0.1×10^6 CAR T-cells/mouse. As shown in Supplemental Figure 4A, the tumor bioluminescence is comparable between the emapalumab-treated and untreated animals.

3. Along the same lines, it would be relevant that in the context of IFN γ blockade, that *in vivo* T cell proliferation actually occurs.

We agree with the Reviewer on this matter, and, as recommended, we have now included the data on the *in vivo* expansion of CAR T-cells (Supplemental Figure 3B-D). Many thanks for this helpful suggestion.

4. I'm not sure whether irradiated tumors are appropriate for the humanized mouse model wherein the antigen load would be more relevant. CRS is seen most commonly in the context of high tumor burden. This is not the case with the presented CRS/ICANS mouse model presented. What happens in the context of high tumor burden (more clinically relevant), when mice are treated with CAR T cells? Does IFN γ blockade still prevent CRS?

We thank the Reviewer for the comment that allow us to better clarify our CRS model with irradiated tumor cells. As reported also in the reply to Reviewer #1, high tumor burden is one of the prerequisites for the onset of CRS. However, generation of a mouse model with a highly proliferating tumor burden could result in the onset of parallel development of sufferance or death for tumor-related toxicity. The choice of using irradiated tumor cells is aligned to the Reviewer comment. Please, also refer to the answer to the Q2 of Reviewer #1.

5. Brain hemorrhages seen is not classical for ICANS, are there other markers for ICANS which can be demonstrated in this model to be more compelling? Have the authors looked at other markers for HLH such as coagulopathy, ferritin etc?

Cerebral edema and brain hemorrhages have been observed in patients with ICANS (doi: 10.1158/2159-8290.CD-17-0698), and we were also able to clearly document them in the *in vivo* CRS model of humanized mice. To better characterize our model, we have now included a molecular signature to document modulation in the glial profile of cerebral tissues of mice treated with emapalumab. With respect to the evaluation of additional marker for coagulopathy and inflammation in our CRS murine model, we have performed an additional animal study, but, unfortunately, we had technical issues to perform biochemistry evaluation in the peripheral blood samples of the included mice due to the low blood volume that was possible to obtain.

Reviewer #3 (Remarks to the Author):

The manuscript submitted by Manni et al. aims to address the mechanisms underlying CRS and neurotoxicity common side effects observed in patients with B-cell malignancies after CAR-T cell

infusion. The authors use *in vitro* and *in vivo* models to assess the role of IFN-gamma in these side effects by using an FDA-approved blocking monoclonal antibody, emapalumab. Even if this is a highly relevant topic in the field and the concept may be of interest, I consider that the study is quite underdeveloped at this stage. More robust experimental data should be provided to conclude that blocking IFN-gamma decreases CAR T-cell-induced side effects while maintaining unaffected anti-cancer efficacy. Also, the study at its present form has some important deficiencies.

First, a deeper characterization of the effects of IFN-gamma blocking on both CAR-T and target cells is missing. In Fig.1 it is not clear to me why authors measure target cell killing at 7 days of co-culture, what does happen at shorter timepoints (e.g. 16 or 24 h)?. Kinetics of cell death, state of apoptotic mediators, etc. should be evaluated.

We thank the Reviewer for these comments. In the new version of the manuscript, we have included data obtained from a new *in vitro* co-culture assay between CAR.CD19 T-cells and Daudi tumor cells (Figure 1D and Figure 2D) either in the presence or in the absence of 100ug/ml emapalumab. This assay was performed by the Incucyte Live Cell Analysis System, a real-time, live cell quantitative imaging platform that allows to make time-lapsed, kinetic measurements from living cells over hours up to days, thereby providing insight into active biological killing kinetics in real time. As suggested by the Reviewer, the tumor elimination kinetics has been shown for very early time points (2 hrs) up to 3 days. In our experience, it is also useful to consider a long-term assay to measure the anti-tumor activity of CAR-T cells, since a sub-optimal tumor elimination will lead to a significant re-expansion of the tumor cells over 7 days of monitoring of the culture. For this reason, we wanted also to include in our paper, data from a long-term co-culture assay, as the data provided in Figure 1A, 1C, Supplemental Figure 1A and 1C, and Figure 2A.

Fig.2 aims to characterize the effects of IFN-gamma blocking in CAR-T cells, however these results could not be evaluated due to poor quality/resolution. In addition, a broader characterization of T cell activation should be evaluated at the protein level, not only measuring CD25 and CD69 activation markers (Suppl Fig. 2).

We thank the Reviewer to underline the need to better characterize T cell activation. In the new version of the paper, we reported the data in Supplemental Figure2 and Figure 3 with a higher resolution and the enlarged activation panel for the cytofluorimetric analysis, including CD25, CD40L, HLA-DR, CD28, CD38, CD44 and CD69 T-cell activation markers after 16 hours of co-

culture with Daudi tumor cells. As reported in Figure 4, the activation profile of CAR T-cells was not modulated by the IFN γ neutralization.

It is not clear to me why CAR-T cell activation was measured with Fc-CD19 instead of using target lymphoma cells, which would provide more valuable information.

We thank the Reviewer for offering us the opportunity to further clarify this point. Nanostring analysis is a technology that allows the study of gene expression on mRNA samples. Therefore, the presence of CAR.CD19 T-cells and Daudi tumor cells in the co-culture assays would not have allowed to be completely ensure about the mRNA extraction from the pure cell population of interest (namely, CAR.CD19 T cells). Moreover, we also agree with the Reviewer that the targeting of CD19+ lymphoma cells will give a valuable information. In light of this consideration, the activation profile evaluated by FACS analysis has been performed after CAR T-cell exposure to Daudi cells (Figure 4).

In Fig. 3 anti-cancer efficacy is determined by injecting CAR-T cells two days after inoculating lymphoma cells, while first signs of detectable tumor burden are observed at day 14. A therapeutic model with detectable tumor burden should be used to resemble the situation in patients. Moreover, emapalumab is administrated 7 days after CAR-T injection, which does not seem to be the best setting to test if IFN-gamma blocking does not affect anti-cancer efficacy. Emapalumab should be rather administrated concomitantly with CAR-T cells, similarly to the following settings aiming to assess CAR-T side effects (Figs. 4-6).

We thank the Reviewer for these valuable comments and suggestions. In the revised version of the manuscript, we performed a new experiment by applying a schedule of treatment based on the infusion of emapalumab at the same time of CAR T-cells as suggested by the Reviewer. As shown in Figure 5E, mice were *i.v.* engrafted with Daudi FF-LUC cells on day -8 to obtain a tumor burden around 10^7 bioluminescence at day 0 (1 log bioluminescence higher compared to the previous setting reported in Figure 5A); after tumor engraftment, mice received *i.v.* the control NT or 10×10^6 CAR.CD19 T-cells, followed by the *i.p.* administration of either 100 mg/kg emapalumab or vehicle on Days 0, 3 and 6. Also in this challenging model suggested by the Reviewer and characterized by a high tumor burden, CAR.CD19 T-cells exerted a significant lymphoma control (Figure 5E, mice #13 to #16; Figure 5F-D) , which did not differ from that observed in mice treated without co-administration of emapalumab (Figure 5E, mice #9 to #12; Figure 5F-D).

The study also lacks a characterization of CAR-T cells in vivo after IFN-gamma blocking. Is this affecting CAR-T cell expansion?

We thank the Reviewer for this comment. As previously discussed also in the reply to the Reviewer #1 and Reviewer #2 comments, in the new version of the paper, we have included the results concerning the expansion of CAR T-cells both in the presence and in the absence of IFN γ neutralization. As shown in Supplementary Figures 3B-D, in the NSG model developed to prove the anti-lymphoma activity of CAR T-cells, the effector T-cell expansion is unaffected by IFN γ neutralization (Supplementary Figure 3A). In the context of the humanized model, although it is difficult to observe CAR T-cell expansion given the short-lived nature of the model, the percentage of CAR T-cells in the peripheral blood of mice is comparable between the untreated cohort and the emapalumab-treated mice (Supplementary Figure 7).

Finally, the evidence supporting that IFN-gamma blocking reduces CAR-T cell side effects is not convincing. In particular, using only hematoxylin/eosin to assess multifocal haemorrhagic lesions and/or macrophage activating syndrome in bone marrow seems limited.

As recommended by the Reviewer, we have now included a deeper evaluation of the glial signaling modulation by performing gene expression analysis in the brain tissues of mice developing CRS, treated or not with emapalumab (Figure 8). Concerning the macrophage activating syndrome, besides the bone marrow histological findings, we also reported that the IFN γ neutralization resulted into a significant decrease of the CXCL-9 chemokines, a molecule that has been shown to be highly correlated with HLH/MAS activity (Locatelli F. et al. *N Engl J Med* 2020; 382:1811-1822; Jacquemin, P. et al. *Br J Clin Pharmacol* 2022).

In addition, the manuscript does not clearly state how many animals are being analyzed in each experiment and how many times each experiment was repeated. Also, it is not clear which statistics analyses were performed in each case.

We thank the Reviewer for these remarks. As recommended, we have now carefully reported in the figure legends the number of animals for each included experiment. Moreover, the statistics analysis applied in each experiment has been reported in the figure legends, as well as in the Materials and Methods section.

I would recommend addressing the above issues prior to consider submitting this manuscript for publication.

REVIEWER COMMENTS

Reviewer #1 (Remarks to the Author):

the authors have addressed my concerns

Reviewer #2 (Remarks to the Author):

The revised manuscript is suitable for publication

Reviewer #3 (Remarks to the Author):

The revised version of the manuscript submitted by Manni et al. has not addressed all the concerns raised during the first revision. I still consider that more robust experimental data should be provided to conclude that blocking IFN-gamma decreases CAR T-cell-induced side effects while maintaining unaffected anti-cancer efficacy. In general, the manuscript remains sloppy in terms of data presentation, number of replicates, statistical comparisons, etc.

The authors do not observe target cell killing at short (e.g. 6, 16 or 24 h) timepoints, which is standard for cytotoxicity assays using CAR-T cells. The authors did not measure cell death or apoptotic mediators. The authors included a proliferation assay for CAR-T cells, however, there are no quantifications of these assays (statistics, replicates, etc.). The authors evaluated different activation markers of CAR-T cells co-cultured with target Daudi cells. However, there is no mention of which of this marker upregulation was statistically significant, numbers of replicates and independent experiments performed. The authors present a therapeutic model administrating CAR-T cells to mice with detectable tumor burden with emapalumab administrated concomitantly with CAR-T cells. However, the studies meant to measure CAR-T cell expansion are difficult to interpret since, why are they quantifying percentage of CAR-T+ cells instead of frequencies of CD45+ cells or absolute numbers? The evidence supporting that IFN-gamma blocking reduces CAR-T cell toxicity remains not convincing because the limited experimental approaches and reduced number of specimens analyzed. This is a key part of the manuscript.

Overall, I think the quality of the manuscript is till poor and the data does not fully work support the conclusions and claims.

Point-by-Point responses to the Reviewers

Please, see in yellow the Author's responses.

Reviewer #1 (Remarks to the Author):

the authors have addressed my concerns

We thank the Reviewer for the appreciation of our efforts to address all the concerns raised during the first revision.

Reviewer #2 (Remarks to the Author):

The revised manuscript is suitable for publication

We thank the Reviewer for considering our revised manuscript suitable for publication on *Nature Communications Journal*

Reviewer #3 (Remarks to the Author):

The revised version of the manuscript submitted by Manni et al. has not addressed all the concerns raised during the first revision. I still consider that more robust experimental data should be provided to conclude that blocking IFN-gamma decreases CAR T-cell-induced side effects while maintaining unaffected anti-cancer efficacy. In general, the manuscript remains sloppy in terms of data presentation, number of replicates, statistical comparisons, etc.

We are sorry for not having fully satisfied the Reviewer's concerns despite our great efforts to address all the points raised during the first review. The Reviewer will find below the answers/clarifications to all the observations raised in this second evaluation.

The authors do not observe target cell killing at short (e.g. 6, 16 or 24 h) time points, which is standard for cytotoxicity assays using CAR-T cells.

As requested by the Reviewer in the first revision, we had included in the re-submitted version of the paper an assay able to monitor over time the target cell killing, from an early time-point up to a very late time-point. Indeed, in order to clearly show the possible impact of emapalumab on tumor cell killing, we had developed a sophisticated assay, based on the use of Incucyte. This experiment has been able to show the cell killing in real-time, from 2hrs up to 64hrs of co-culture between target and effector cells, with an evaluation of residual GFP+ cells performed every 2hrs. This assay clearly showed the kinetics of the target elimination, helping to reveal any possible difference in cell death induced by different experimental conditions. With this assay, we were able to show that IFN γ inhibition does not affect the kinetics of CAR.CD19 T-cell activity.

In order to further address the Reviewer point to focus on a very early time point, we have now performed a standard 6-hour cytotoxic assay, using three independent donors, based on a sensitive assay able to evaluate the cytotoxic activity of CAR T cells at a very early time point (6hrs). Data have now been provided in Supplemental Figure 1D. As clearly shown, the presence of emapalumab does not impact on the cytotoxicity of CAR T cells against Raji cell line.

The authors did not measure cell death or apoptotic mediators.

In response to the Reviewer comment, we would like to specify that the readout in each assay shown in Figure 1A, 1C, 1D, Supplemental 1A, Supplemental 1C, Figure 2A and 2C all refers to a measure of cell death of target leukaemia cells induced by CAR T cells, either in the presence or in the absence of emapalumab. Indeed,

these figures are all showing the residual, alive, tumor cells after the co-culture with CAR.CD19 T cells, as explained in the material methods and in the legends of each specific figure.

To address the specific request of measure of apoptosis, we have now included novel data in Supplemental Figure 1E, showing that, at an early time point of co-culture with CAR.CD19 T cells (6 hrs), we do not observe any difference in the levels of early apoptotic Raji cells (evaluated by flow-cytometry, as Annexin V⁺/7-AAD⁻ cells) upon exposure to emapalumab. The experiments have been carried out using three independent CAR.CD19 T cell donors.

The authors included a proliferation assay for CAR-T cells, however, there are no quantifications of these assays (statistics, replicates, etc.).

We thank the Reviewer for this observation. The data included in Figure 1E and 2D clearly show a complete overlap of the proliferating CAR T cells upon stimulation with the target cells, either in the presence or in the absence of emapalumab. The experiment was performed on three independent CAR.CD19 T cell donors, obtaining overlapping results and a p value with none statistical significance. For this reason, we have shown the flow-cytometry read-out of one representative donor for CAR.CD19.41bb (Figure 1E) and for CAR.CD19.28 T cells (2D). Based on the Reviewer request, we have now specified in the Figure legends that the assay was performed on three donors, and that the p value, calculated using the t-student test, was >0.05 between CAR.CD19 T-cells stimulated in the presence or absence of emapalumab.

The authors evaluated different activation markers of CAR-T cells co-cultured with target Daudi cells. However, there is no mention of which of this marker upregulation was statistically significant, numbers of replicates and independent experiments performed.

Three independent experiments with three different CAR.CD19 T cell donors were performed. As recommended by the Reviewer, we have now specified these details in the Figure legend. All the reported activation markers showed a significant modulation when CAR T cells were activated by target cells either in the presence or in the absence of emapalumab, as compared to untransduced T cells (also exposed to the target cell line DAUDI). In our previous version of the manuscript, in order to provide more clarity on the evidence of absence of modulation of the level of expression of activation markers on CAR T cells upon addition of emapalumab, we had not reported the statistical significance of the modulation in comparison to untransduced T cells. As requested by the Reviewer, we have now included in the figure the statistical significance of the differences observed between NT T cells and CAR T cells.

The authors present a therapeutic model administering CAR-T cells to mice with detectable tumor burden with emapalumab administered concomitantly with CAR-T cells. However, the studies meant to measure CAR-T cell expansion are difficult to interpret since, why are they quantifying percentage of CAR-T⁺ cells instead of frequencies of CD45⁺ cells or absolute numbers?

We would like to point out that the choice to quantify the percentage of CAR T cells and not of CD45⁺ cells in our animal model derives from the need to be very accurate in providing data in the manuscript. Indeed, due to the high leukaemia burden, the evaluation of CD45⁺ cells could be misleading, considering that leukaemia cells are also CD45⁺.

However, to address the request of the Reviewer, we have now included a novel Supplemental Figure 3B, reporting the percentage of total CD45⁺ GFP⁻ cells in the animals included in the experiment. Considering that the leukaemia cells engrafted in the animals were GFP⁺, this evaluation is able to exclude tumour cells and to refer to CAR T cells only.

The evidence supporting that IFN-gamma blocking reduces CAR-T cell toxicity remains not convincing because the limited experimental approaches and reduced number of specimens analyzed. This is a key part of the manuscript.

Overall, I think the quality of the manuscript is till poor and the data does not fully work support the conclusions and claims.

We respectfully disagree with this Reviewer comment, because we sincerely think that our data robustly prove, both *in vitro* and *in vivo*, that the IFN- γ blockade does not reduce the activity of CAR.CD19 T cells in the setting of both leukaemia and lymphoma, and that this approach can be of high value to control toxicity of CAR T cells. Moreover, our data are in line with other papers, already published and reported in our references, and we strongly believe that the publication of this paper in Nature Communications can represent a valuable reference for all the clinical approaches that will be developed soon.

REVIEWERS' COMMENTS

Reviewer #3 (Remarks to the Author):

The authors have now addressed my concerns. I would suggest that graphs showed individual values.

Point-by-Point responses to the Reviewers

Please, see in yellow the Author's responses.

Reviewer #3 (Remarks to the Author):

The authors have now addressed my concerns. I would suggest that graphs showed individual values.

We are pleased to have addressed the reviewer's concerns, whose suggestions helped us to improve the merit of the manuscript. We have also modified all the graphs showing individual values as suggested.